# Natural Products from Medicinal Plants with Anti-Human Coronavirus Activities

**DOI:** 10.3390/molecules26061754

**Published:** 2021-03-21

**Authors:** Salar Hafez Ghoran, Mohamed El-Shazly, Nazim Sekeroglu, Anake Kijjoa

**Affiliations:** 1Department of Chemistry, Faculty of Science, Golestan University, 15759-49138 Gorgan, Iran; 2Department of Pharmaceutical Biology, Faculty of Pharmacy and Biotechnology, The German University in Cairo, 11835 Cairo, Egypt; mohamed.elshazly@pharma.asu.edu.eg; 3Department of Horticulture, Faculty of Agriculture, Kilis 7 Aralik University, 79000 Kilis, Turkey; nsekeroglu@gmail.com; 4ICBAS-Instituto de Ciências Biomédicas Abel Salazar and CIIMAR, Rua de Jorge Viterbo Ferreira, 228, 4050-313 Porto, Portugal

**Keywords:** COVID-19, SARS-CoV, phytochemicals, medicinal plants, ACE2, enzyme inhibitors

## Abstract

Since the emergence of severe acute respiratory syndrome caused by coronavirus 2 (SARS-CoV-2) first reported in Wuhan, China in December 2019, COVID-19 has spread to all the continents at an unprecedented pace. This pandemic has caused not only hundreds of thousands of mortalities but also a huge economic setback throughout the world. Therefore, the scientific communities around the world have focused on finding antiviral therapeutic agents to either fight or halt the spread of SARS-CoV-2. Since certain medicinal plants and herbal formulae have proved to be effective in treatment of similar viral infections such as those caused by SARS and Ebola, scientists have paid more attention to natural products for effective treatment of this devastating pandemic. This review summarizes studies and ethnobotanical information on plants and their constituents used for treatment of infections caused by viruses related to the coronavirus family. Herein, we provide a critical analysis of previous reports and how to exploit published data for the discovery of novel therapeutic leads to fight against COVID-19.

## 1. Introduction

On 12 February 2020, the emergence of a severe respiratory infectious disease was announced by the World Health Organization (WHO), known as Human Coronavirus Diseases 2019 (COVID-19) [1]. In the following weeks, thorough phylogenetic investigations revealed the exact identity of the virus and its similarity to the previous severe acute respiratory syndrome coronavirus (SARS-CoV) with 89.1% nucleotide similarity and also more than 95% homology with key drug targets. The new lethal and highly infectious virus was designated as SARS-like coronavirus type 2 (SARS-CoV-2) [2,3]. To date, four human coronaviruses (HCoV) have been identified viz. HCoV-OC43; β-CoV, HCoV-229E; α-CoV, HCoV-NL63, and HCoV-HKU1 that are responsible for SARS-CoV diseases [4]. The fact that SARS-CoV-2 has high phylogenic similarity to HCoV-OC43 and HCoV-229E, the main causative agents of the common cold, has guided scientists in their search for new anti-SARS-CoV-2 compounds [5]. Moreover, it was found that the angiotensin-converting enzyme 2 (ACE2) in the SARS-CoV-2 host receptor is the same as that in the SARS-CoV host receptor. Therefore, targeting ACE2 can be one of potentially promising approaches to hamper the current pneumonia outbreak [6,7,8]. In addition, previous studies highlighted the importance of using drugs to inhibit enzymes involved in SARS-CoV replication such as SARS-CoV 3C-like protease (SARS-CoV 3CL^Pro^) and papain-like protease (SARS-CoV PL^Pro^). These enzyme inhibitors stop the virus from expressing key replicative enzymes such as RNA-dependent RNA polymerase (RdRp) and helicase (Figure 1) [9,10,11].

On the other hand, pathological findings of COVID-19 revealed that approximately 20% of patients are suffering from an acute respiratory distress syndrome (ARDS), which is sparked by a “cytokine storm”, causing an uncontrolled inflammation. This phenomenon is associated with various inflammation-related cytokines, chemokines such as interleukins (IL)-1β, IL-6, IL-7, IL-8, IL-9, IL-17, tumor necrosis factor (TNF)-α, interferon (IFN)-γ, granulocyte-colony stimulating factor (G-CSF), granulocyte-macrophage colony-stimulating factor (GM-CSF), and chemokine ligands 2, 3, and 4 (CCL2-CCL4), which have been observed in plasma obtained from individuals suffering from COVID-19 and even worse in dying patients [12]. It is worth mentioning that even interferons (IFNs), including IFN-α, -β, and -γ, have been recognized as potential agents for treatment of SARS. Cinatl et al., in their experiment with recombinant interferons toward two clinical strains of SARS-CoV-FFM-1 in Vero and Caco2 cells, have found that IFN-α significantly inhibited the replication of SARS-CoV, however, its selectivity index (SI) was 50–90 times lower than that of IFN-β. Despite the inactivity of IFN-γ in Caco2 cell cultures, it showed a better activity than IFN-α in Vero cell cultures [13]. 

The increasing daily infections and mortality rates caused by COVID-19, combined with the lack of effective medicines, call for researchers’ urgent attempts to tackle this serious problem. One of the approaches, besides anti-inflammation and immune-nanomedicine strategies [12,14], is focusing on medicinal plants and their secondary metabolites since it is well recognized that they have played a crucial role in the discovery of potential anti-infective agents. One of the most efficient strategies in counteracting SARS-CoV could be the use of natural products that are able to prevent viral adhesion. These compounds potentially interact with the receptor-mediated recognition and consequently block the interaction process between viruses and host cells. Another alternative could be enhancing the immune system by taking various kinds of plant products and their related preparations like dietary supplements [15,16]. The importance of natural products to fight against SARS-CoV is reflected in a review article by Kumar et al. that describes synthetic and naturally occurring compounds of different scaffolds that were patented from 2008–2013 for their anti-SARS-CoV activities with different mechanisms of action. Natural products patented in this period as SARS-CoV 3CL^pro^ inhibitors include kaurane diterpenes pseurata A (**1**) and liangshanin A (**2**), flavonols quercetagenin (**3**) and robinetin (**4**), and pentacyclic triterpenoid quinone methides celatrol (**5**) and tingenone (**6**) (Figure 2) [17]. 

The initial outbreak of COVID-19 in Wuhan quickly became a worldwide pandemic and caused hundreds of thousands of lives and economic setbacks throughout the world. Due to the genetic similarity of SARS-CoV-2 and the previously emerged SARS-CoV viz. CoV HCoVNL63, HCoV-HKU1 and Middle East respiratory syndrome coronavirus (MERS-CoV), this review describes the exploration of some medicinal plants, most of which are used in traditional medicine, and their secondary metabolites for treatment of the diseases caused by SARS-CoV. The search engines used for this review are Web of Science, Scopus, PubMed, and Google scholar. The secondary metabolites cover a diverse array of structural classes. Overall, the review discusses the structures and mechanisms underlying their activities of ca. one hundred naturally occurring phytochemicals with anti-SARS-CoV activities reported to date.

## 2. Medicinal Plants with Anti-HCoV Properties

The development of new antiviral drugs is an urgent issue to pursue because of life-threatening viral diseases such as Ebola, SARS, and MERS. Many plants have produced phytochemicals with great potential to combat these diseases. For example, Traditional Chinese Medicine (TCM) specialists have described *Toona sinensis* Reom (also known as *Cedrela sinensis*, Family Meliaceae), a tree commonly found in Taiwan, China, and Malaysia that produced various phenolic compounds and sterols, for its beneficial effects against HCoV-229E. The tender leaf extract of this plant showed in vitro inhibitory effects toward SARS-CoV replication with SI of 12–17 [18]. In vitro antiviral assays of extracts of some medicinal plants including *Cimicifuga racemosa* (L.) Nutt. (Ranunculaceae), *Coptis chinensis* Franch. (Ranunculaceae), *Melia azedarach* L. (Meliaceae), *Phellodendron amurense* Rupr. (Rutaceae), *Sophora subprostrata* Chun and T.Chen. (Fabaceae), and *Paeonia suffruticosa* Andrews (Paeoniaceae) have revealed their potential as anti-SARS-CoV candidates for further clinical studies [19]. Due to the deficiency of therapeutic medicines, *Houttuynia cordata* Thunb. (Saururaceae) was indicated to deal with SARS symptoms because of its traditional use to relieve lung-related abnormalities such as lung abscess, mucus, cough, and pneumonia [20]. Experimental results showed that the aqueous extract of *H. cordata* increased the proliferation of CD4^+^ and CD8^+^ lymphocytic T cells, whereas the levels of inflammation cytokines, IL-2 and IL-10, were effectively ameliorated. SARS-CoV 3CL^Pro^ activity was also inhibited by administration of the *H. cordata* extract [21,22]. Since out of 200 Chinese medicinal herbs including *Lycoris radiata* Herb. (Amaryllidaceae), *Artemisia annua* L. (Asteraceae), *Pyrrosia lingua* (Thunb.) Farw. (Ploypodiaceae), and *Lindera aggregata* (Sims) Kostem (Lauraceae) have been documented for their moderate to potent antiviral activities against SARS-CoV, with half maximal effective concentrations (EC_50_) ranging from 2.4 to 88.2 μg/mL, these plants have been proposed for anti-SARS-CoV remedies [23]. Another plant with various antiviral properties is *Echinacea purpurea* (L.) Moench (Asteraceae), whose preparation is also known as Echinaforce^®^, which was evaluated for its in vitro anti-SARS-CoV-2 activity [24]. The results showed that treatment with this preparation was able to irreversibly inhibit HCoV-229E activity with a half-maximal inhibitory concentration (IC_50_) of 3.2 μg/mL. Interestingly, this preparation is also effective against other highly pathogenic CoVs, such as those causing SARS and MERS [25]. It is important to mention also that chamomile (*Anthemis hyalina* DC.) flowers, black cumin (*Nigella sativa* L.) seeds, and *Citrus sinensis* L. Osbeck (Rutaceae) peels are commonly used as herbal remedies in traditional medicines in many cultures for treatment of a variety of human diseases. Ulasli et al., in their evaluation of the effects of the ethanol extracts of these three medicinal plants on the CoV replication and the expression of the transient receptor potential (TRP) genes during CoV infection, have found that treatment of HeLa-CEACAM1a (HeLa-epithelial carcinoembryonic antigen-related cell adhesion molecule 1a) cells with these plant extracts prior to infection with MHV-A59 (mouse hepatitis virus-A59) decreased the replication of the virus. Although all the extracts had an effect on IL-8 secretion, TRP gene expression and viral load after coronavirus infection, it was the flower extract of *Anthemis hyalina* DC. that showed the most significant difference in viral load [26]. A survey of medicinal plants on SARS and SARS-like infectious diseases revealed that *Scutellaria baicalensis* Georgi, *Bupleurum chinense* DC., and *Gardenia jasminoides* J. Ellis. could significantly prevent acute infection in SARS patients [27]. Following the immunotherapy approach, Sultan et al. proposed that garlic (*Allium sativum* L. fam.), ginger (*Zingiber officinalis*), *Hypericum perforatum* L., *Camellia sinensis* (L.) Kuntze, *N. sativa* L., and liquorice (*Glycyrrhiza glabra* L.) could be used for COVID-19 prevention since they are well-known immune enhancer plants [28]. Other examples of medicinal herbs that have potential for treatment or alleviate COVID-19 symptoms are presented in Table 1.

## 3. Plant Secondary Metabolites with Anti-HCoV Properties

For practicality, the anti-HCoV compounds are categorized according to their chemical classes while the mechanisms of actions and their docking studies are also discussed for each class of compounds when available.

### 3.1. Alkaloids

During a screening for antiviral activities of Chinese medicinal herb extracts against SARS-CoV, using MTS (3-(4,5-dimethylthiazol-2-yl)-5-(3-carboxymethoxyphenyl)-2-(4-sulfophenyl)-2H-tetrazolium inner salt) assay for virus-induced cytopathic effect (CPE), Li et al. have found that the extract of *Lycoris radiata* Herb. showed the most potent effect with EC_50_ value of 2.4 µg/mL. Chemical investigation of the active fraction resulted in the isolation of the alkaloid lycorine (**7**) (Figure 3). Compound **7** displayed a strong inhibitory effect against SARS-CoV with EC_50_ value of 15.7 nM and SI more than 900. It is worth mentioning that the authors have found that the isolated compound was more potent against SARS-CoV than the commercially available lycorine (EC_50_ = 48.8 nM) (Table 2) [23].

*Bis*-benzylisoquinoline alkaloids tetrandrine (**8**), fangchinoline (**9**), and cepharanthine (**10**) (Figure 3), isolated from *Stephania tetrandra* S. Moore and other members of the family Menispermaceae, also displayed anti-HCoV activities. Since HCoV-OC43 is not only most closely related to SARS-CoV and MERS-CoV but also shares several functional properties with both of them, Kim et al. used HCoV-OC43-infected MRC-5 fibroblasts, derived from human lung tissue, as a model to evaluate antiviral activities of **8-10** and to elucidate their underlying mechanisms. They found that treatment with **8-10** Gdose-dependently increased cell viability against HCoV-OC43 infection. Compound **8** displayed the most potent antiviral activity with an IC_50_ value of 295.6 nM, whereas **9** and **10** were less active presenting IC_50_ values of 919.2 and 729.7 nM, respectively. Therefore, the SIs of **8**–**10** were higher than 40, 11, and 13, respectively. It was also found that anti-HCoV-OC43 activities of **8**–**10** were dose- and time-dependent and that they are most effective at the early stage of the HCoV-OC43 life cycle, and the timing of treatment is more important to protect against virus-induced cell death than the duration of cell exposure to the compounds. Compounds **8**–**10** also inhibited the replication of HCoV-OC43 as well as the expression of the nucleocapsid (N) and spike (S) proteins (Table 2) [30].

By using a cell-based assay, with SARS-CoV and Vero E6 cells, for a screening of existing drugs, natural products, and synthetic compounds to identify effective anti-SARS agents, Wu et al. have found that among more than 10,000 agents screened, around 50 compounds were active at concentration of 10 µM, among which was the existing drug reserpine (**11**) (Figure 4), an indole alkaloid produced by several members of *Rauwolfia* spp. (Apocynaceae). Compound **11** displayed an EC_50_ of 3.4 µM, and a SI of 7.3. Since **11** has been used clinically, the authors speculated that its related natural products may also be active against SARS-CoV (Table 2). By using the International Species Information System (ISIS) database to search for commercially available compounds whose structures have 80% similarity to **11**, they have found that six compounds, i.e., **12**–**17** (Figure 4), were related to **11**. However, **12**–**14** displayed the minimal concentration of inhibition toward SARS-CoV much higher than reserpine (**11**), ranging from 10–20 µM, whereas the values for **15**–**17** were ca. 100 µM [31].

### 3.2. Anthraquinones 

Targeting the interaction of SARS-CoV spike (S) protein with ACE2 is of particular interest in the search for anti-SARS-CoV agents. By screening Chinese medicinal herbs, supervised by the Committee on Chinese Medicine and Pharmacy in Taiwan, Ho et al. have found that three widely used Chinese medicinal herbs viz. root tubers of *Rheum officinale* Baill. and *Polygonum multiflorum* Thunb., vines of *P. multiflorum* Thunb., all belonging to the family Polygonaceae, inhibited the interaction of S protein with ACE2 of SARS-CoV with IC_50_ values ranging from 1 to 10 μg/mL. Since emodin (**18**) (Figure 5) is a major constituent of these herbs, this compound was evaluated for the interaction of S protein with ACE2. The results showed that **18** blocked the binding of S protein to ACE2, in a dose-dependent manner, with an IC_50_ value of 200 µM. Compound **18** also blocked the binding of S protein of SARS-CoV to Vero E6 cells. When compared to promazine (the positive control), the percent inhibition of **18** was 94.12 ± 5.90% at 50 µM, whereas that of promazine was 95.6 ± 7.7% at 5 µM. Hence, the anti-SARS-CoV activity of both compounds was not due to their toxicity [32]. Compound **18** was also shown to inhibit an ion channel 3a protein, encoded by an open reading frame ORF-3a, in the infected cells whose activity may influence virus release [45]. The ORF-3a is also known as “New gene” localized between “spike and envelope gene” (SNE), and has been identified in other coronaviruses including the SNE of HCoV-OC43 that shows similar ion-channel characteristics to the 3a protein of SARS-CoV [46]. This new observation, together with the finding that **18** may disrupt the interaction of S protein and ACE2 [32], supports the suggestion that **18** or its derivatives may become potent new therapeutic agents for treatment of SARS and other coronavirus-induced diseases.

### 3.3. Flavonoids and Flavonoid Glycosides

During the investigation of the phenolic-containing Chinese herbs used for the prevention of SARS during its outbreaks in China, Lin et al. examined the anti-SARS-CoV 3CL^pro^ effect of extracts of some Chinese herbs and herb-derived phenolic compounds using a cell-free cleavage and cell-based cleavage assay. The results showed that a flavanone hesperetin (**19**) (Figure 5) dose-dependently inhibited the SARS-CoV 3CL^Pro^ cleavage activity with an IC_50_ value of 8.3 μM. However, the capacity of **19** to inhibit SARS-CoV-2 replication has never been investigated (Table 2) [47]. On the other hand, baicalin (**20**) (Figure 5), a flavone glycoside isolated from a Chinese medicinal plant *Scutellaria* spp. (Lamiaceae), was found to display an antiviral activity against the prototype strains (39849) of SARS-CoV in foetal rhesus kidney-4 (fRhK-4) cell line with an EC_50_ value of 12.5 µg/mL in 48 h, and against 10 strains of SARS-CoV with EC_50_ values ranging from 12.5-25 µg/mL in 48 h, with SI values higher than 4 to 8 [48]. Both **20** [33] and scutellarin (**21**) (Figure 5) showed a neuroprotective effect in addition to a disruption of the interaction of S protein with ACE2 [49]. Luteolin (**22**) and quercetin (**23**) (Figure 5), naturally occurring flavones, were also found to exhibit anti-SARS-CoV activity through avid interaction with the surface S protein of a wild-type SARS-CoV, and subsequently interfering the viral entry into Vero E6 cells with EC_50_ values of 10.6 and 83.4 μM, respectively (Table 2) [50].

Another potential target is a viral helicase. Since this enzyme is essential for viral genome replication, it is currently considered as a potential target for antiviral drug development. Thus, in an attempt to find natural compounds with inhibitory activity against SARS-CoV, Yu et al. [51] have evaluated a series of naturally occurring compounds, including flavonoids, xanthones, terpenoids, sterols and fatty acids, against the activity of SARS helicase, nsP13, using fluorescence resonance energy transfer (FRET)-based double-strand (ds) DNA unwinding assay or a colorimetry-based ATP hydrolysis assay. Interestingly, only myricetin (**24**) and scutellarein (**25**) (Figure 5) were found to inhibit the ATPase activity of nsP13 by more than 90% at a concentration of 10 µM, with IC_50_ values of 2.71 and 0.86 µM, respectively. The toxicity of **24** and **25** against normal breast epithelial MCF10A cells were also examined at a concentration of 2 μM and it was found that both compounds are safe at pharmacologically-effective concentrations [51].

It is well documented that naringin (**26**) (Figure 5), a constituent of citrus fruits, exhibits a potent anti-inflammatory activity by inhibiting the expression of the pro-inflammatory cytokines such as COX-2, iNOS, IL-1β and IL-6, induced by LPS in vitro. Since **26** could also inhibit high mobility group box protein 1 (HMGB1) expression, it could restrain cytokine storm by preventing up-regulation of cytokines such as TNF-α, IL-6, IL-1β, and IL-8 [52]. Based on this observation, an attempt to identify effective antiviral and anti-inflammation compounds from *Citrus* flavonoids has been made to give a nutritional recommendation for the prevention and treatment of COVID-19. Some Citrus flavonoids including naringenin (**27**), hesperidin (**28**) (Figure 5), neohesperidin (**29**), nobiletin (**30**) (Figure 6), together with vitamin C have found to be potential SARS-CoV-2 inhibitors whose mechanisms involved ACE2 receptor inhibition and reduction of intracellular oxidative stress triggered by inflammation and virus infection [53,54,55,56]. In addition, **27** not only inhibited both HCoV-OC43 and HCoV-229E in Vero E6 cells at a concentration of 62.5 μM, but also showed an inhibitory behavior in cytopathic effect (CPE) at 72 h post-infection along with a protection of cells from SARS-CoV-2 infection at concentrations of 250 and 62.5 μM [54].

Jo et al. used a synthetic peptide labelled with an Edans-Daabcyl FRET pair to search for SARS-CoV 3CL^pro^ inhibitors against a flavonoids library consisting of ten different scaffolds. Among 64 compounds, tested at 20 µM, only herbacetin (**31**), rhoifolin (**32**) and pectolinarin (**33**) (Figure 6) were found to exhibit prominent inhibitory activity, with IC_50_ values of 33.17, 27.45 and 37.78 µM, respectively, at a concentration ranging from 2 to 320 µM. To avoid a non-specific inhibition of these flavonoids caused by aggregation through complexity, the assays in the presence of 0.01% Triton X-100 were performed resulting in a better inhibitory activity than those without using Triton X-100. Since SARS-CoV 3CL^pro^ contains one tryptophan residue (Trp-31) at the catalytic domain, a general tryptophan-based assay method was used to independently confirm the inhibitory activity of the three flavonoids. As the three flavonoids decreased the emission intensity, the authors concluded that there was a complex formation between the catalytic domain and these flavonoids. The interaction between SARS-CoV 3CL^pro^ and the three inhibitory flavonoids were analyzed to predict the binding affinities by induced-fit docking study. The docking study of kaempferol (**34**) and morin (**35**) (Figure 6) were also performed and compared with herbacetin (**31**) to find structural features that confer good affinity of **31**. The results showed that these compounds share the kaempferol (**34**) motif. The docking study further revealed that the phenyl moiety of **34** occupies the S1 site of SARS-CoV through a hydrogen bond with Glu-166, whereas its chromen-4-one scaffold locates in the S2 site. For **31**, there are four hydrogen bonds formed within a distance of 2.33 Å in the S2 site. However, the major bonding force was due to the presence of the hydroxyl group on C-8 that binds with Glu-166 and Gln-189. These bindings were predicted to confer a good glide score (−9.263) to **31**. Due to the lack of the hydroxyl group on C-8, **34** and **35** have two hydrogen bonds less than **31**. On the other hand, the binding modes of **32** and **33** are different from those of **34** and **31**. Since **32** and **33** possess α-l-rhamnopyranosyl-β-d-glucopyranoside and α-l-mannopyranosyl-β-d-glucopyranoside at C-7 of the chromen-4-one scaffold, there is no hydrogen bond between the hydroxyl group on C-7 and the backbone of Ile-188. Moreover, as the bulky disaccharide groups require a large space to accommodate, they occupy the S1 and S2 sites and the chromen-4-one moieties that locate in the S2 and S3’ sites. Therefore, the better affinity of **32** may be due to a concomitant binding through S1, S2 and S3’ sites (Table 2) [57]. 

By using molecular docking studies and surface plasmon resonance (SPR)/FRET-based assays, Chen et al. have found that quercetin-3-β-d-galactoside (**36**) (Figure 7) was a potent inhibitor of SARS-CoV 3CL^pro^, exhibiting more than 50% inhibition at a concentration of 50 µM (IC_50_ = 42.79 µM in a competitive kinetic mode). Molecular modeling and Q189A mutation experiment revealed that Gln-189 plays a key role in the binding between **36** and SARS-CoV 3CL^pro^. In order to elucidate structure-activity relationship features, the authors have synthesized eight quercetin-3-β-d-galactoside derivatives (**37**–**44**) (Figure 7), and tested their inhibitory activities against SARS-CoV 3CL^pro^ at a concentration of 50 µM. The results revealed that (i)-elimination of hydroxyl groups on the quercetin moiety (**42**–**44**) significantly diminished inhibitory activity, (ii)-acetoxylation of the sugar moiety (**37**) decreased an inhibitory activity of the compound, and (iii)-introduction of a large sugar moiety to the hydroxyl group on C-7 of quercetin (**23**) did not cause any adverse effects on the ability of the inhibitor to functionally disrupt the SARS-CoV 3CL^pro^ activity (Table 2) [58].

Twelve geranylated flavonoids, including tomentins A–E (**45–49**), 3′-*O*-methyldiplacol (**50**), 4′-*O*-methyldiplacol (**51**), 3′-*O*-methyldiplacone (**52**), 4′-*O*-methyldiplacone (**53**), mimulone (**54**), diplacone (**55**) and 6-geranyl-4′,5,7-trihydroxy-3′,5′-dimethoxyflavone (**56**) (Figure 8) were isolated from the methanolic extract of fruits of a Chinese medicinal plant, *Paulownia tomentosa* (Thunb.) Steud. (Paulowniaceae). Compounds **45–56** inhibited papain-like protease (PL^pro^) activity in a reversible, mixed-type mode, with IC_50_ values ranging from 5.0–14.4 µM (Table 2) [34].

In a continuation of their work on emodin (**18**), Schwarz et al. have investigated whether kaempferol (**34**) and its glycosides can block the 3a channel protein of coronavirus by determination of their efficacy to inhibit Ba^2+^-sensitive current. They have found that **34**, at 20 µM, was able to reduce endogenous Ba^2+^-sensitive current, and the degree of inhibition was independent on the voltage. In contrast to **18**, which selectively inhibited the 3a-mediated current and at 20 µM already produced more than 50% block, **34** exhibited similar effects against the endogenous and the 3a-mediated components. Juglanin (**57**) (Figure 9) seemed to be the most potent kaempferol glycoside that showed complete inhibition at 20 µM. Interestingly, the IC_50_ of **57** could be obtained even at a concentration as low as 2.3 µM, indicating that **57** is about one order of magnitude more potent to block 3a-protein channel than **18**. Although afzelin (**58**) and tiliroside (**59**) (Figure 9) were less potent than **57**, they were as effective as **18**. Compound **59**, at 20 µM, produced a block to 0.48 ± 0.09, whereas **57**, at the same concentration, completely blocked the 3a-mediated current. Interestingly, only 10 µM of **58** (inhibition to 0.83 ± 0.01) showed a similar degree of inhibition to 20 µM of **34**. Curiously, the acylated kaempferol derivatives, kaempferol-3-*O*-(2,6-di-*p*-coumaroyl)-glucopyranoside (**60**) and kaempferol-3-*O*-(3,4-diacetyl-2,6-di-*p*-coumaroyl) glucopyranoside (**61**) (Figure 9), both containing an additional *p*-coumaroyl group, showed no effect on Ba^2+^-sensitive current at 20 µM, whereas the kaempferol triglycoside kaempferol-3-*O*-α-l-rhamnopyranosyl(1→2)[α-l-rhamnopyranosyl(1→6)]-β-d-glucopyranoside (**62**) (Figure 9) exhibited about 30% inhibition at 20 µM, which is similar to the effect of **58** at 20 µM. Of note is that both compounds contain rhamnose residues. It is interesting to note also that **23**, **27** and the isoflavone genistein (**63**) (Figure 9), all of which are known for their antiviral potency including SARS CoV, did not exhibit any significant modulation of the 3a-mediated current [59].

As part of an investigation to search for botanical sources for SARS-CoV 3CL^pro^ inhibitors, Ryu et al. have found that biflavones, named amentoflavone (**65**), bilobetin (**66**), ginkgetin (**67**), and sciadopitysin (**68**) (Figure 10), isolated from a medicinal plant *Torreya nucifera* L. Siebold and Zucc. (Lamiaceae), exhibited an inhibitory effect against SARS-CoV 3CL^pro^. However, **65** was the most potent non-competitive anti-SARS-CoV 3CL^pro^ with an IC_50_ value of 8.3 μM. Compound **65** is 30-fold more potent as SARS-CoV 3CL^pro^ inhibitor than the parent compound apigenin (**64**) (Figure 9) (IC_50_ = 280.8 µM), whereas **22** and **23**, whose IC_50_ values were 23.8 and 20.2 µM, respectively, were less potent than **65**. Furthermore, the authors have also attempted to elucidate the interaction of SARS-CoV 3CL^pro^ with **65** by using a three-dimensional structure of SARS-CoV 3CL^pro^ in complex with a substrate-analogue inhibitor (coded 2z3e)24, obtained from the Protein Data Bank for modeling analysis. Computer docking analysis revealed that **65** nicely fits in the binding pocket of 3CL^pro^. The hydroxyl group on C-5 of **65** formed two hydrogen bonds with the nitrogen atom of the imidazole ring of His-163 (3.154 Å) and a hydroxyl group of Leu-141 (2.966 Å), both belonging to S1 site of 3CL^pro^, whereas the hydroxyl group in the B ring of **65** forms hydrogen bonds with Gln-189 (3.033 Å), belonging to S2 site of 3CL^pro^. These structure-activity relationships studies implied that interactions with Val-186 (4.228 Å) and Gln-192 (3.898 Å) are one of the key chemotypes in this inhibitor. Furthermore, the potencies of **65** and **66** correlated well with their binding energies viz **65** = 11.42 kcal/mol, and **64** = 7.79 kcal/mol. The difference in binding energy could explain the reason why the IC_50_ value of **65** against 3CL^pro^ is 30-fold smaller than that of **64** (Table 2) [35].

### 3.4. Lignans and Neolignans

The dibenzylbutyrolactone lignans hinokinin (**69**) and savinin (**70**), isolated from the ethyl acetate extract of the heartwood of *Chamaecyparis obtusa* var. *formosana*, a synthetic phenyltetralin lignan 4,4′-*O*-benzoylisolariciresinol (**71**), and two naturally occurring biphenyl neolignans honokiol (**72**) and magnolol (**73**) (Figure 11) were assayed for their anti-SARS-CoV activities using a cell-based assay to measure a SARS-CoV-induced CPE on Vero E6 cells [36]. Compounds **69**–**73** exhibited strong inhibitory activity in the CPE assays at concentrations between 3.3 and 20 µM. Interestingly, **69** exhibited significant inhibitory effects at concentrations as low as 1 µM. The anti-SARS-CoV replication activity, measured by an ELISA method, revealed that among the lignans and neolignans tested, **70** exhibited the strongest activity, with an EC_50_ value of 1.13 µM, which is lower than that of the reference compound valinomycin (EC_50_ = 1.63 µM), whereas **69** showed the weakest activity (EC_50_ > 10 µM). The neolignans **72** and **73** displayed significant activity with EC_50_ values ranging from 3.8 to 7.5 µM. Moreover, **70** also showed inhibitory activity against SARS-CoV 3CL^pro^ with an IC_50_ value of 25 µM whereas the IC_50_ of **69** was more than 100 µM (Table 2) [36].

### 3.5. Gallic Acid Derivatives

By using a two-step screening method which combines a frontal affinity chromatography-mass spectrometry (FAC/MS) and pseudotyped virus infection assay to search for drugs that can interfere with the entry of SARS-CoV into host cells, Yi et al. have found that, among small molecule libraries consisting of extracts from 121 Chinese herbs, tetra-*O*-galloyl-β-d-glucose (TGG, **74**) (Figure 12) exhibited potent antiviral activities against the human immunodeficiency virus (HIV)-luc/SARS pseudotyped virus, with an EC_50_ value of 2.86 µM. Interestingly, **74** showed little anti-vesicular stomatitis virus (VSV) activity at the same concentration levels that can effectively inhibit the entry of HIV-luc/SARS pseudotyped virus to its host cells, indicating that **74** is highly specific against SARS-CoV. Since **74** was isolated through analysis of its binding to the S2 protein of SARS-CoV, this compound most likely blocks the entry of HIV-luc/SARS pseudotyped virus to its host cells. Moreover, **74** also inhibited, in a dose-dependent manner, wild-type SARS-CoV infection with an EC_50_ value of 4.5 µM (Table 2) [60].

In a screening of a panel of compounds with a capacity to inhibit SARS-3CL^pro^ activity by HPLC assay using a protease inhibitor, *N*-ethylmaleimide, as positive control, Chen et al. have found that tannic acid (**75**) (Figure 12) and theaflavin-3′-gallate (TF2B, **76**) were active against 3CL^pro^. Fluorogenic substrate peptide assay revealed that **75**, **76** and theaflavin-3,3′-digallate (TF3, **77**) (Figure 13), abundant in black tea, displayed relevant anti-SARS-3CL^pro^ activity with IC_50_ values less than 10 µM (Table 2) [61].

### 3.6. Terpenoids 

#### 3.6.1. Monoterpenoids

Monoterpenes are major constituents of essential oils. Loizzo et al. have evaluated essential oils from various members of the Cupressaceae family for their inhibitory activity against SARS-CoV and Herpes simplex virus type 1 (HSV-1) replication in vitro by visually scoring of the virus induced-cytopathogenic effect post-infection. They have found that the berries oil of *Laurus nobilis* L. (Lauraceae), whose major constituents are β-ocimene (**78**), 1,8-cineole (**79**), α-pinene (**80**), and β-pinene (**81**) (Figure 14), exhibited an IC_50_ value of 120 µg/mL against SARS-CoV, with SI = 4.2, whereas *Thuja orientalis* L. and *Juniperus oxycedrus* ssp. *oxycedrus* oils displayed weaker activities with IC_50_ values of 130 and 270 µg/mL, and SI = 3.8 and 3.7, respectively, when compared with glycyrrhizin (**111)**, as the positive control. Gas chromatography/mass spectrometry (GC/MS) analysis revealed that **77** and δ-3-carene (**82**) (Figure 14) are major constituents of the *T. orientalis* oil whereas **80** and β-myrcene (**83**) are predominant in the berries oil of *J. oxycedrus* ssp. *oxycedrus* [38]. Since many essential oils are used in aromatherapy, this approach could be promising in preventing SARS.

#### 3.6.2. Sesquiterpenoids

In the evaluation of phytochemicals for SARS-CoV activity, Wen et al. have isolated two sesquiterpenes, cedrane-3β,12-diol (**84**) and α-cardinol (**85**) (Figure 14) from the ethyl acetate extract of the heartwood of *Juniperus formosana* Hayata (Cupressaceae). Compounds **84** and **85** exhibited inhibition of CPE of SARS-CoV on Vero E6 cells at concentrations as low as 3.3 and 1 µM, respectively. Moreover, **85** inhibited SARS-CoV replication with an EC_50_ of 4.44 µM (SI = 17.3) (Table 2) [36]. Interestingly, a sesquiterpene α-cedrol (**86**) (Figure 14), also a major component of *T. orientalis* L. essential oil, showed an inhibitory activity against SARS-CoV with IC_50_ of 130 ± 0.4 μg/mL and SI = 4.2, when compared to **111** (the positive control; IC_50_ of 641.0 μg/mL (779 μM), SI = 1.2) [38].

#### 3.6.3. Diterpenoids

Eight abietane and two labdane diterpenoids comprising ferruginol (**87**), dehydroabieta-7-one (**88**), sugiol (**89**), 8β-hydroxyabieta-9(11),13-dien-12-one (**91**), 6,7-dehydroroyleanone (**93**), pinusolidic acid (**95**) (isolated from the ethyl acetate extract of the heartwood of *C. obtusa* var. *formosana*), cryptojaponol (**90**) and 7β-hydroxydeoxycryptojaponol (**92**) (isolated from the heartwood of *Cryptomeria japonica*) and 3β,12-diacetoxyabieta-6,8,11,13-tetraene (**91**) (isolated from the ethyl acetate extract of the heartwood of *J. formosana* Hayata), together with foskalin (**96**) (Figure 15), were evaluated for their anti-SARS-CoV activities using a cell-based assay to measure SARS-CoV-induced CPE on Vero E6 cells [36]. Eight abietane (**87**–**94**) and two labdane diterpenes (**95** and **96**) showed strong inhibitory activity in the CPE assays at concentrations ranging from 3.3 to 20 µM. Although the SI of the five abietane-type diterpenes **87**, **88**, **91**, **92**, and **94** and two labdane-type diterpenes **95** and **96** were 58, 76.3, >510, 111, 193, >159, 89.8, none of the diterpenoids tested inhibited SARS-CoV 3CL^pro^ at concentrations of less than 100 µM. In comparison with the positive controls, niclosamide and valinomycin, most of the diterpenoids with potent activity against CPE also exhibited marked inhibitory effects on SARS-CoV replication. The SI values for **87**, **88**, **91**, **92**, **94**–**96** are substantially higher than that of the positive control valinomycin (SI = 41.4). The authors, therefore, proposed that the abietane diterpenes **87**–**94** possessed potent anti-SARS-CoV activities, however, these activities apparently did not involve an inhibition of SARS-CoV 3CL^pro^ (Table 2) [36].

Six abietane diterpenes, 18-hydroxyferruginol (**97**), hinokiol (**98**), ferruginol (**87**), 18-oxoferruginol (**99**), *O*-acetyl-18-hydroxyferruginol (**100**), methyl dehydroabietate (**101**), one primarane diterpene, isopimaric acid (**102**), and one labdane diterpene, kayadiol (**103**) (Figure 16), isolated from the *n*-hexane fraction of leaves of *T. nucifera* L. Siebold and Zucc. (Lamiaceae), were investigated for their inhibitory activities against SARS-CoV 3CL^pro^ by using a FRET assay. Compounds **97**–**103** displayed inhibition against SARS-CoV 3CL^pro^ at concentrations up to 100 µM, whereas **89** exhibited significantly higher inhibitory effects on 3CL^pro^ with an IC_50_ = 49.6 µM. Moreover, **89** was nearly 4-fold more potent than a parent abietane diterpene, abietic acid (**104**) (IC_50_ = 189.1 µM) (Table 2) [35].

#### 3.6.4. Triterpenoids

Betulinic acid (**105**) and betulonic acid (**106**) (Figure 17), isolated from the ethyl acetate extract of the heartwood of *J. formosana* Hayata, also showed strong inhibitory activity in the CPE assays at concentrations between 3.3 and 20 µM. Compound **105** exhibited inhibitory effects on SARS-CoV 3CL^pro^ (IC_50_ = 10 µM) 10-fold stronger than **106** (IC_50_ > 10 µM). Compound **105** showed a competitive inhibition mode of action with *Ki* value of 8.2 ± 0.7 µM. Docking study revealed that **105** fitted nicely into the substrate-binding pocket of SARS-CoV 3CL^pro^. Moreover, the binding was strengthened by a hydrogen bond formed by the hydroxyl group on C-3 of **105** with the oxygen atom of the carbonyl group of Thr-24, located at the *N*-terminus of domain I (residues 8–101) of the 3CL^pro^. On the contrary, **106** showed only hydrophobic interaction but did not form any intermolecular bonds with the enzyme (Table 2) [36].

Chang et al. have evaluated the anti-HCoV-229E activity of 22 triterpenoids including four friedelanes, three glutinanes, one lupane, three taraxeranes, two oleananes, one dammarane, one ocotillone, two simiarenanes, three cycloartanes, and one cycloeucalane, isolated from leaves of *Euphorbia neriifolia* L. (Euphorbiaceae). However, only 3β-friedelanol (**107**), 3β-acetoxyfriedelane (**108**), friedelin (**109**) and epitaraxerol (**110**) (Figure 17) exhibited anti- HCoV-229E activity. Compounds **107–109** exhibited more potent activity than the positive control, actinomycin D. The percent of inhibition were in the following descending order: **107** (132.4%) > **110** (111.0%) > **109** (109.0%) > **108** (80.9%) > (actinomycin D 69.5%). Most significantly, exposure to these compounds led to an improvement in cell viability [39]. 

#### 3.6.5. Saponins

Glycyrrhizin (**111**) (Figure 18), a triterpenoid saponin mainly isolated from roots of *Glycyrrhiza glabra* L. (Fabaceae), is also known to exhibit a myriad of biological activities including antiviral effects against HSV-1, varicella-zoster virus (VZV), hepatitis A (HAV) and B virus (HBV), HIV, Epstein-Barr virus (EBV), human cytomegalovirus and influenza virus, and has been considered as a potential treatment for patients with chronic hepatitis C [62]. Cinatl et al. [62], in their evaluation of the antiviral potential of **111**, ribavirin, 6-azauridine, pyrazofurin and mycophenolic acid against two clinical isolates of coronavirus (FFM-1 and FFM-2) from patients with SARS, have found that **111** was the most potent inhibitor of SARS-CoV replication in Vero cells, with SI = 67, whereas ribavirin and mycophenolic acid did not affect SARS-CoV replication. On the other hand, 6-azauridine and pyrazofurin, inhibitors of orotidine monophosphate decarboxylase, inhibited replication of SARS-CoV at non-toxic doses with SI = 5 and 12, respectively. Compound **111** was also found to inhibit adsorption and penetration of the virus, which are the early steps of the replicative cycle. Moreover, **111** was less effective when added during the adsorption period than when added after virus adsorption (EC_50_ 600 mg/L vs. 2400 mg/L, respectively) and was most effective when given during and after the adsorption period (EC_50_ 300 mg/L) [40]. Due to the efficacy of **111**, Hoever et al. have obtained 15 semisynthetic analogs of glycyrrhizin (**112**–**126**) (Figure 18) and tested them to find more potent compounds against SARS-CoV. They have found that the introduction of *N*-acetylglucosamine into the glycoside chain of glycyrrhizin (**112**) increased the anti-SARS-CoV activity about 9 times when compared to **111**. Compound **112** inhibited SARS-CoV replication at an EC_50_ of 40 µM with SI > 75. Compound **112** showed no CPE at 500 µM. The authors suggested that the introduction of *N*-acetylglucosamine moiety into the carbohydrate portion of **111** increases its hydrophilic property, which might be important for the interaction of glycyrrhizin with viral proteins. Moreover, the authors have speculated that the viral entry is inhibited by binding of *N*-acetylglycosamine to the carbohydrates of the viral S protein. Although the analogs **120**–**123** (Figure 18) were active against SARS-CoV, with EC_50_ values ranging from 5 to 50 µM, they also exhibited high cytotoxicity when compared to **111** and the glycopeptide derivatives **112** and **113**, resulting in low SI values (2 to 5). Compounds **113** and **114**, containing L-Cys(SBn) and Gly-Leu, respectively, were active against SARS-CoV. Compound **113** was 10-fold more active against SARS-CoV than **111**, with an EC_50_ = 35 µM (SI = 41), whereas **114** exhibited an EC_50_ = 139 µM (SI = 2). On the contrary, the glycopeptide analogs **115**–**119** did not exhibit any activity against SARS-CoV in concentrations up to 1 mM. Therefore, it was concluded that a free carboxylic acid group at C-30 of the triterpene scaffold was vital for the anti-SARS-CoV activity of glycyrrhizin glycopeptide analogs since **113** and **114** contain a free carboxylic acid group at C-30 whereas **115**–**119** contain three amino acids or amino acid alkyl ester residues. Interestingly, **125** and **126**, where the two glucuronic acid units were replaced by ortho-hydroxybenzoic acid and succinic acid, respectively, no anti-SARS-CoV activity was observed, indicating that the sugar moiety is essential for the anti-SARS-CoV activity (Table 2) [63].

Another group of oleanane triterpene glycosides is saikosaponins. These compounds were isolated from medicinal plants such as *Bupleurum* spp., *Heteromorpha* spp. and *Scrophularia scorodonia*, and have been reported to possess a myriad of biological activities including antiviral activities against HIV [41], influenza virus [42], varicella-zoster [43], measles and herpes simplex [44]. Cheng et al. have investigated the in vitro antiviral activity and mode of action of saikosaponins A (**127**), D (**128**), B_2_ (**129**) and C (**130**) (Figure 19) against HCoV-229E, which has been recognized as an important cause of nosocomial respiratory viral infections in high-risk infants. It was found that **127-130**, at concentrations of 25 μM or less, significantly inhibited HCoV-229E infection, being **129** the strongest inhibitor (Table 2). Mechanism study revealed that **129** decreased infection by HCoV-229E in a dose- and time-dependent manner, and the inhibition of viral infection was more effective when it was added before viral adsorption than after viral adsorption. This observation suggests that **129** may disturb the early stages of HCV-229E infection, including viral attachment and penetration. Experimental results revealed that **129** significantly inhibited HCoV-229E attachment in a dose-dependent manner, suggesting that **129** prevents the attachment of HCoV-229E into host cells. Compound **129** also inhibited penetration of HCoV-229E into host cells in a time-dependent manner, suggesting that it also prevents the penetration of HCoV-229E into host cells, probably through detachment of virus that has already bound to the cell by disturbing viral glycoproteins [64].

Wu et al. have found that the oleanane type saponin aescin (**131**) (Figure 20), a major active principle of *Aesculus hippocastanum* L. (Sapindaceae), inhibited viral replication in SARS-CoV-infected Vero E6 cells with an EC_50_ of 6.0 µM and CC_50_ of 15 µM (SI = 2.5) [31]. 

In an attempt to decipher the mechanism of action of *N. sativa* L. for its usefulness against SARS-CoV-2, Mani et al. have used a network generated around ACE2, a target site for SARS-CoV-2, to depict a disease network and *N. sativa* L. constituents as a treatment network. Network analysis revealed various hub nodes, i.e., the proteins that control the disperse of signaling in the network including ACE2, which has various functions related to blood-vessel dilation, controlling blood pressure, cytokine production, inflammatory response, and protein transport. The symptoms of breathlessness, dysregulated blood pressure and inflammation are those that were initially reported in COVID-19 diagnosis. Glide Standard-Precision (SP) docking was performed and the best binding ligand to ACE2 was found to be α-hederin (**132**) (Figure 20). Compound **132** (−6.265 Kcal/mol) was observed to be involved in a regulation of blood pressure, cell communication, vascular processes, negative regulation of cell death, response to stress and immune effector processes. The genes identified for regulating the immune response via **132** are ACE2, F2, SRC, etc. Since **132** has ACE as its predicted receptor in human system, this compound can be further explored for targeted therapies [65].

## 4. Future Perspectives and Conclusions

This article presents an overview of the current state of knowledge on the effects of various classes of phytochemicals and their mode of action on SARS-CoV, which can be used as models for drug leads to combat the COVID-19 pandemic. In an attempt to find new anti-HCoV drugs, this review can guide researchers to target medicinal plants with antiviral properties because of their valuable naturally occurring metabolites. We have shown that some structural classes of natural products such as the abietane diterpenes, friedelane and oleanane triterpenes and saponins along with geranylated and glycosylated flavonoids could be considered as unique scaffolds for the development of anti-HCoV-229E drugs. On the other hand, the phenolic compounds such as those containing galloyl moiety can exert antiviral activity through the interaction with the enzyme active sites. Emerging from this review are the glycosylated secondary metabolites, in particular saponins, which have an intrinsic capacity to be used as anti-SARS-CoV diseases. Due to the novel profile of COVID-19 pneumonia related to the viral binding site of ACE2, some secondary metabolites are more likely to represent potential antiviral drugs. Finally, yet importantly, a combination of IFNs, in particular IFN-β, with other antiviral and anti-inflammatory natural products might be of great interest in treatment of SARS-CoV. Further investigations and chemical modifications of anti-SARS-CoV naturally occurring compounds to search for parameters that increase efficacy and decrease toxicity and side effects are needed.

## Figures and Tables

**Figure 1 molecules-26-01754-f001:**
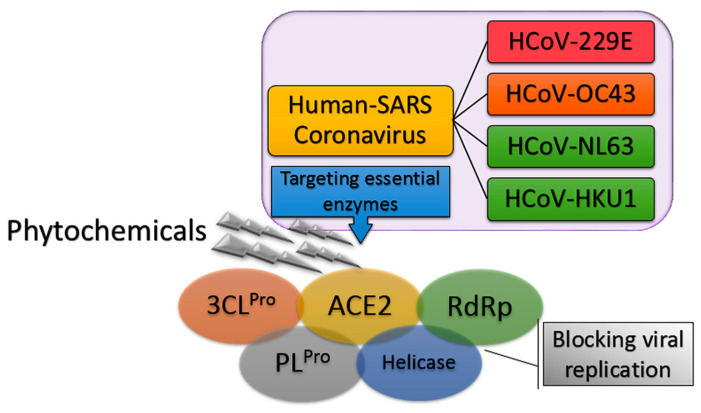
General overview of inhibition of severe acute respiratory syndrome coronavirus (SARS-CoV) replication.

**Figure 2 molecules-26-01754-f002:**
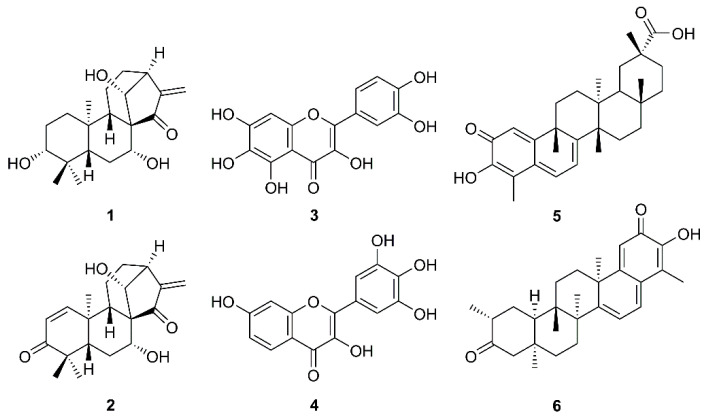
Some natural products patented as anti-SARS-CoV agents from 2008–2013.

**Figure 3 molecules-26-01754-f003:**
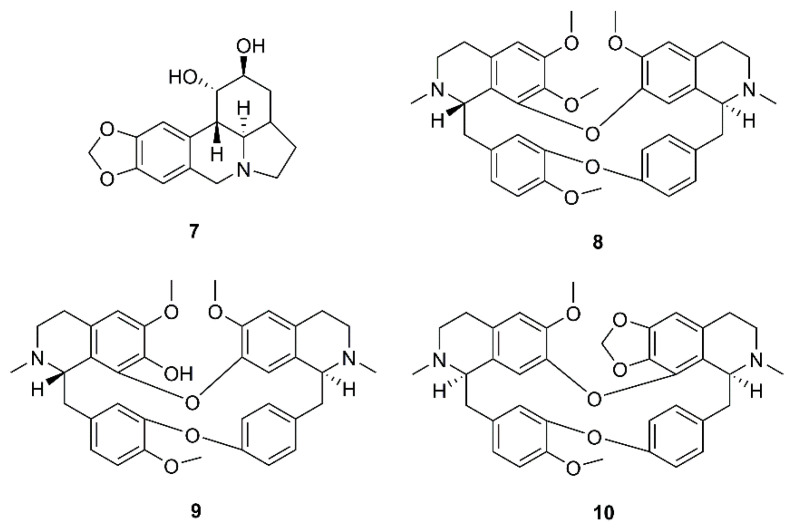
Structures of lycorine (**7**), tetrandrine (**8**), fangchinoline (**9**), and cepharanthine (**10**).

**Figure 4 molecules-26-01754-f004:**
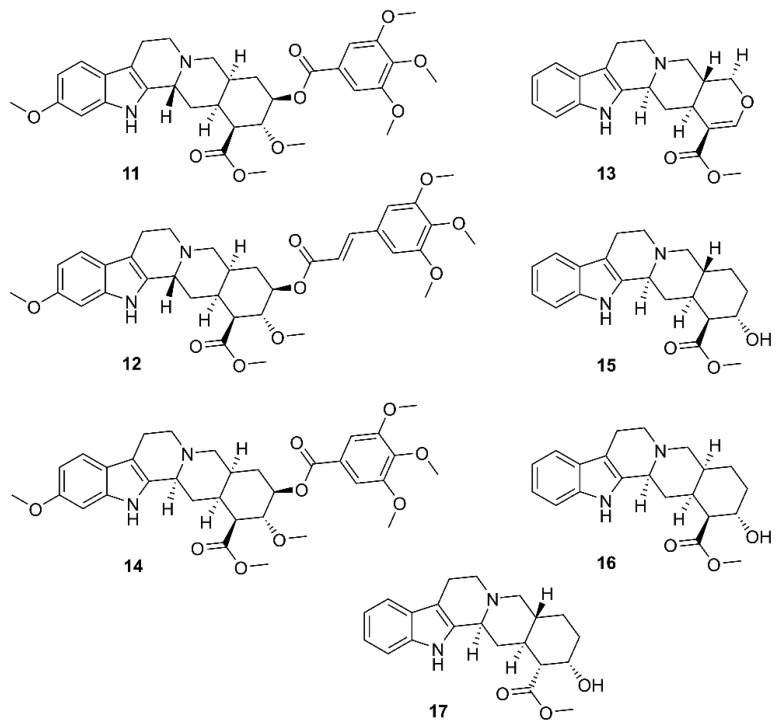
Structures of reserpine (**11**) and its analogs **12**–**17**.

**Figure 5 molecules-26-01754-f005:**
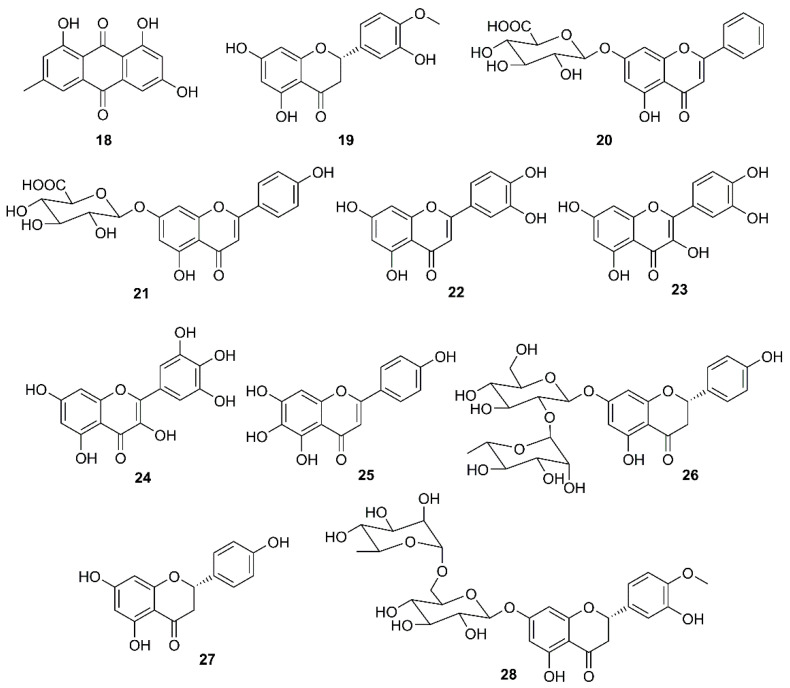
Structures of emodin (**18**), flavonoids, and flavonoids glycosides **19**–**28**.

**Figure 6 molecules-26-01754-f006:**
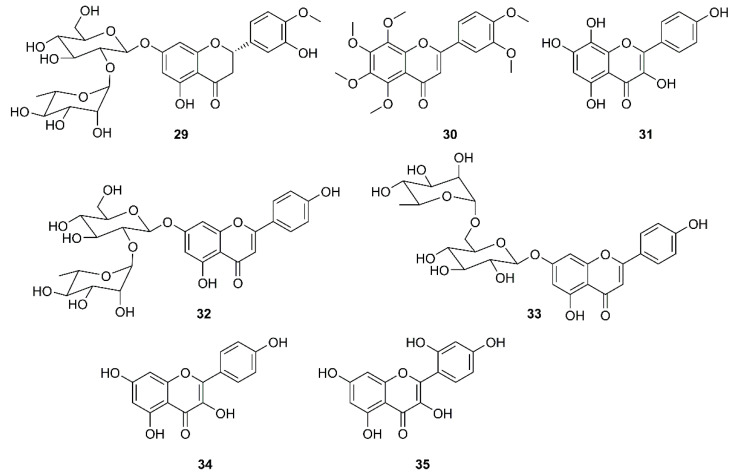
Structures of flavonoids and flavonoid glycosides **29**–**35**.

**Figure 7 molecules-26-01754-f007:**
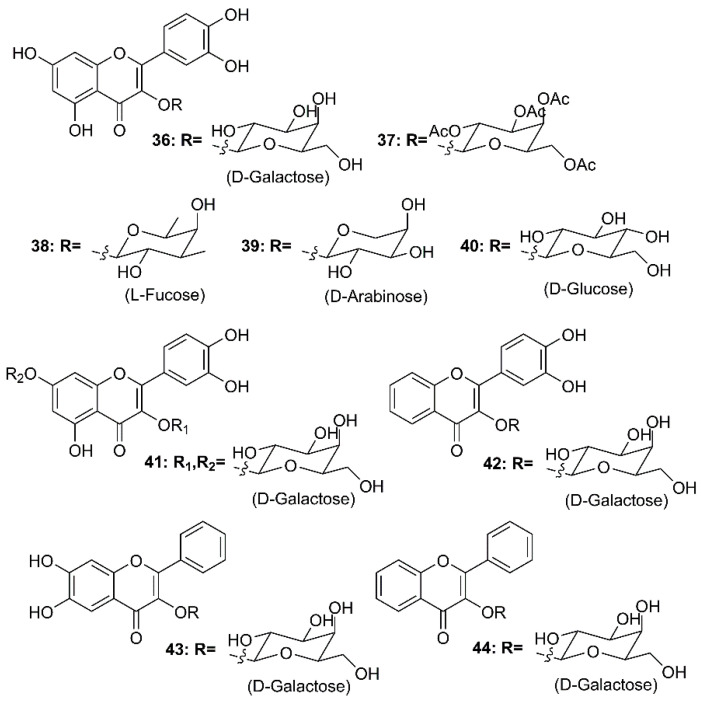
Structures of quercetin-3-β-d-galactoside (**36**) and its synthetic derivatives **37**–**44**.

**Figure 8 molecules-26-01754-f008:**
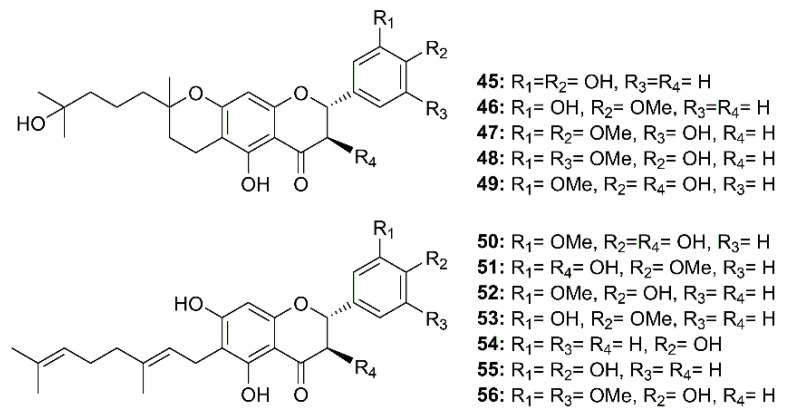
Structure of geranylated flavonoids **45**–**56**.

**Figure 9 molecules-26-01754-f009:**
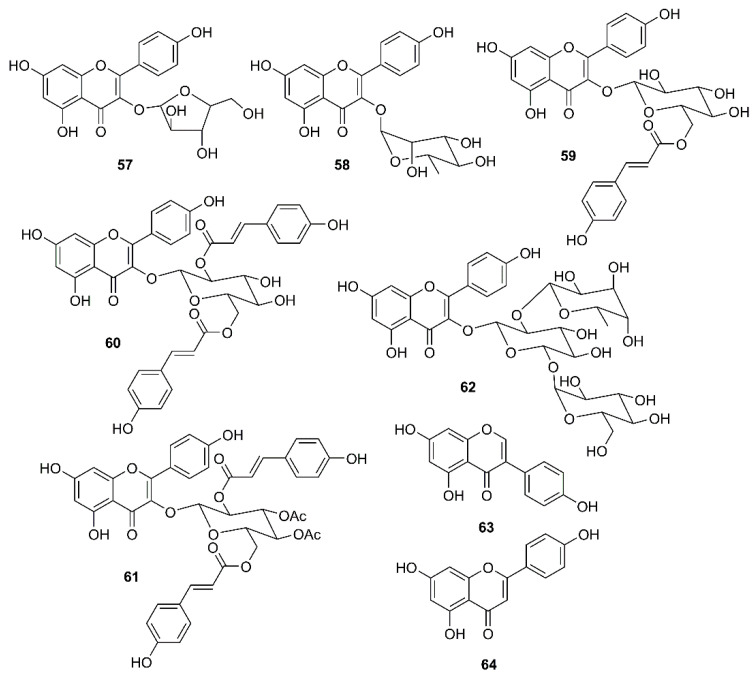
Structure of flavonoids, flavonoid glycosides and an isoflavonoid **57**–**64**.

**Figure 10 molecules-26-01754-f010:**
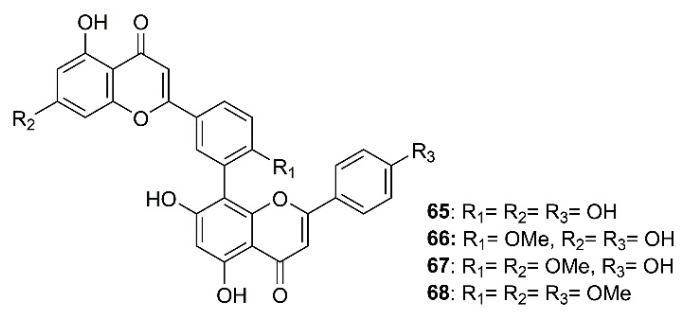
Structure of biflavones **65**–**68**.

**Figure 11 molecules-26-01754-f011:**
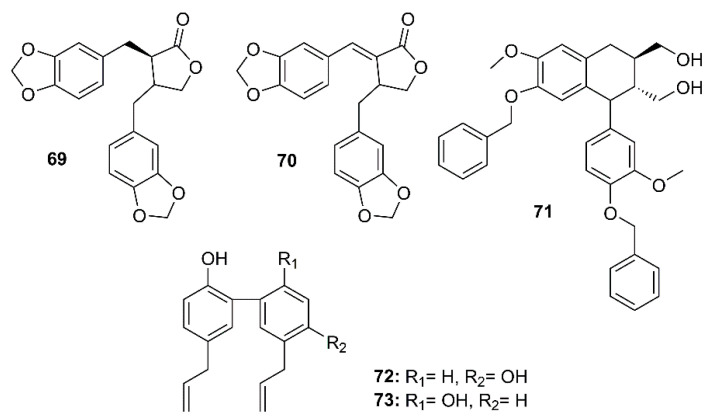
Structures of lignans **69**–**71** and neolignans **72** and **73**.

**Figure 12 molecules-26-01754-f012:**
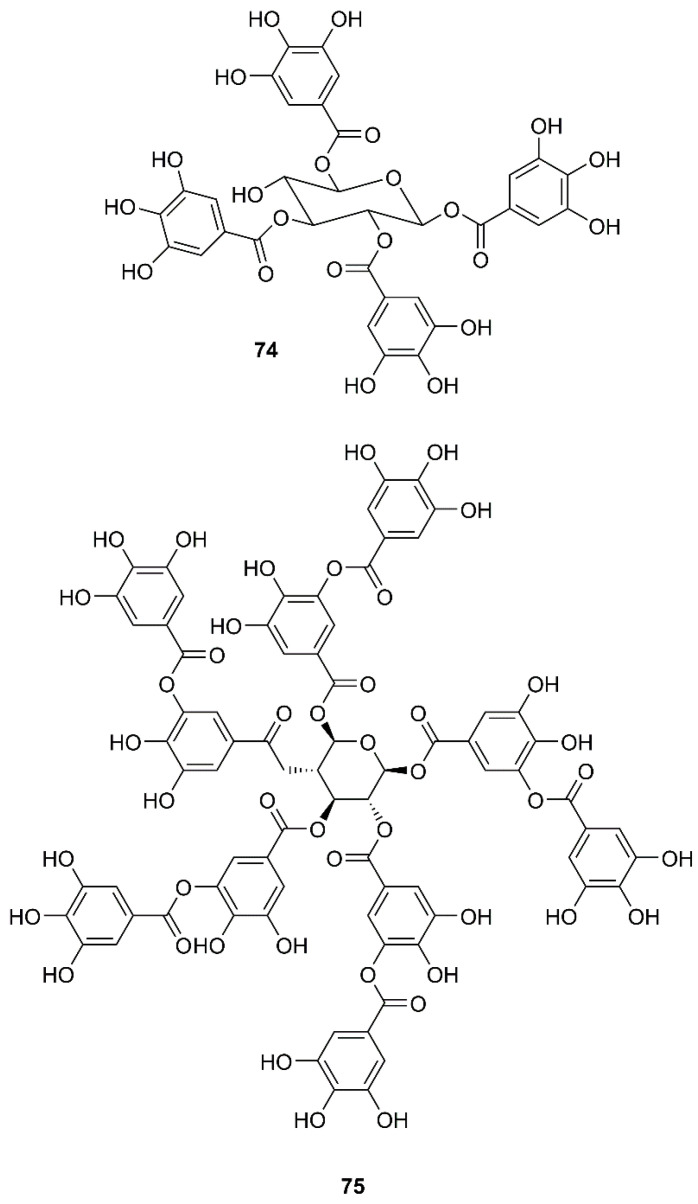
Structures of tetra-*O*-galloyl-β-d-glucose (TGG) (**74**) and tannic acid (**75**).

**Figure 13 molecules-26-01754-f013:**
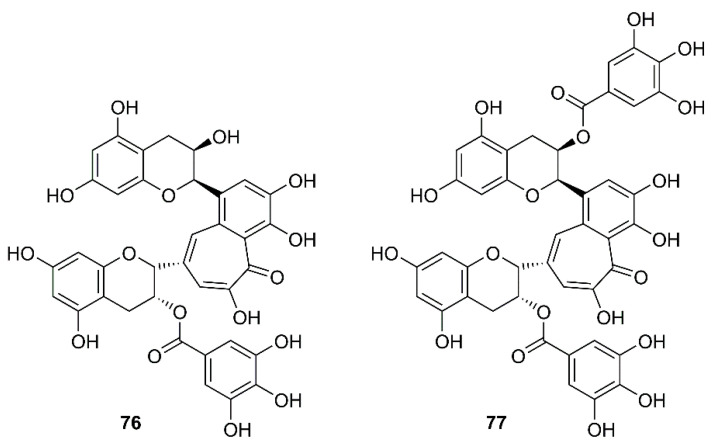
Structures of TF2B (**76**) and TF3 (**77**).

**Figure 14 molecules-26-01754-f014:**
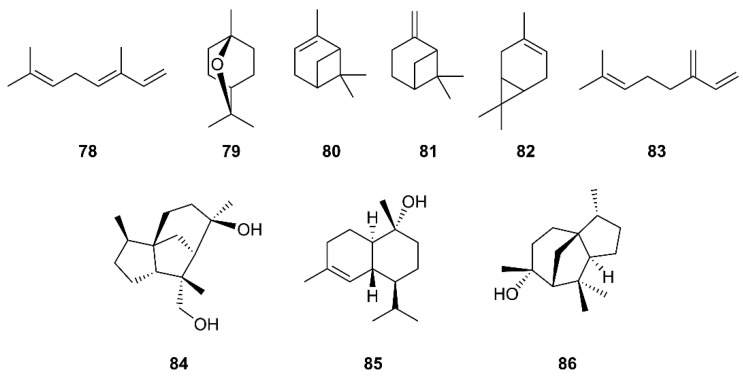
Structures of monoterpenes **78**–**83** and sesquiterpenes **84**–**86**.

**Figure 15 molecules-26-01754-f015:**
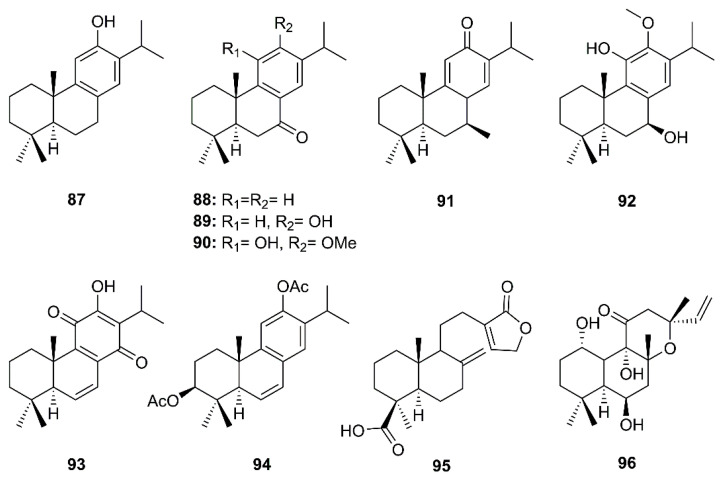
Structures of abietane (**87**–**94)** and labdane (**95** and **96**) diterpenoids.

**Figure 16 molecules-26-01754-f016:**
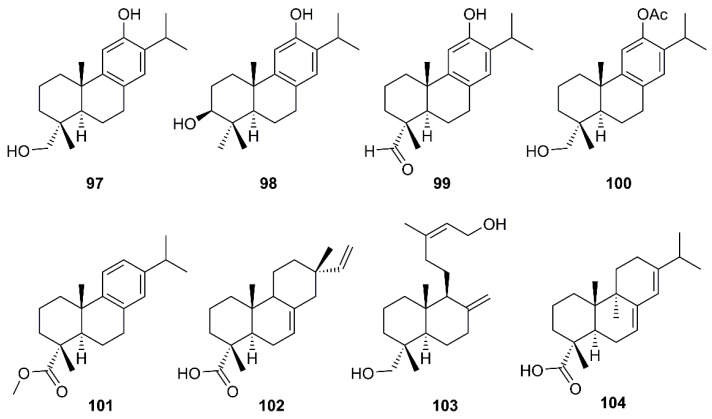
Structures of abietane (**97**–**101** and **104**), primarane (**102**), and labdane (**103**) diterpenoids.

**Figure 17 molecules-26-01754-f017:**
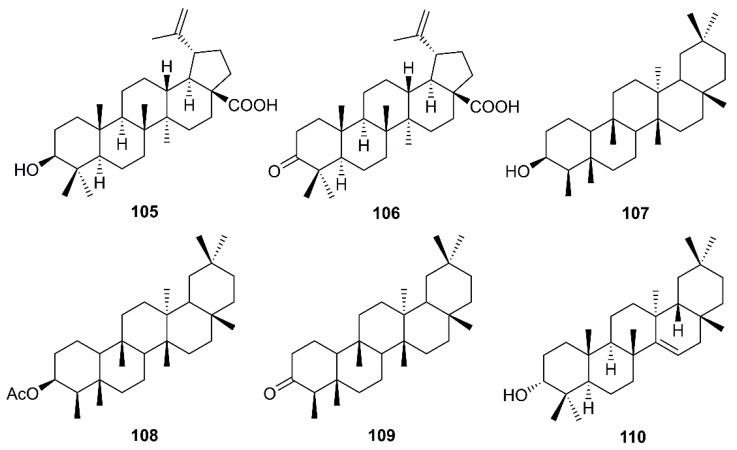
Structures of triterpenoids **105**–**110**.

**Figure 18 molecules-26-01754-f018:**
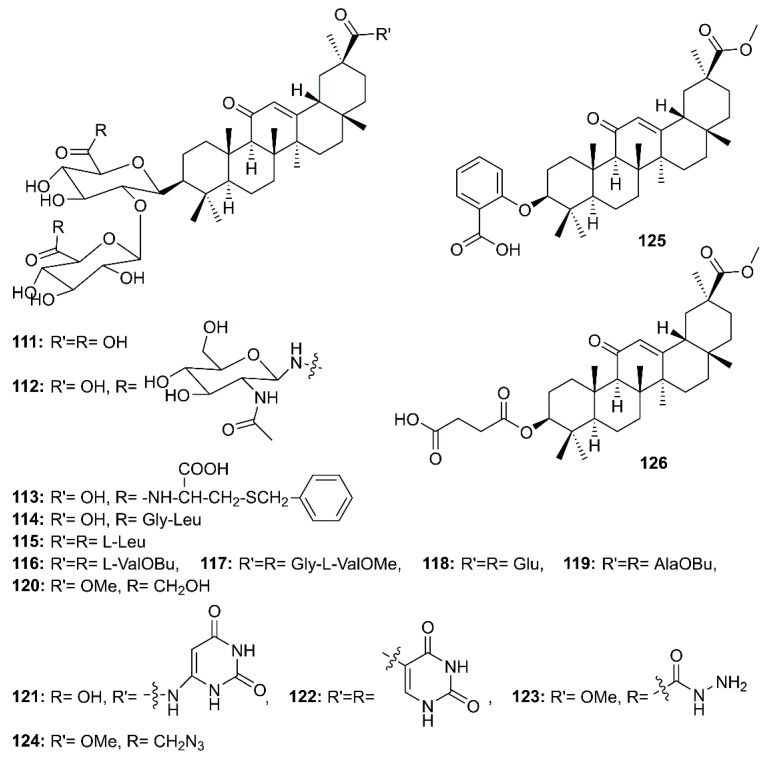
Structures of glycyrrhizin (**111**) and its synthetic analogs **112–126**.

**Figure 19 molecules-26-01754-f019:**
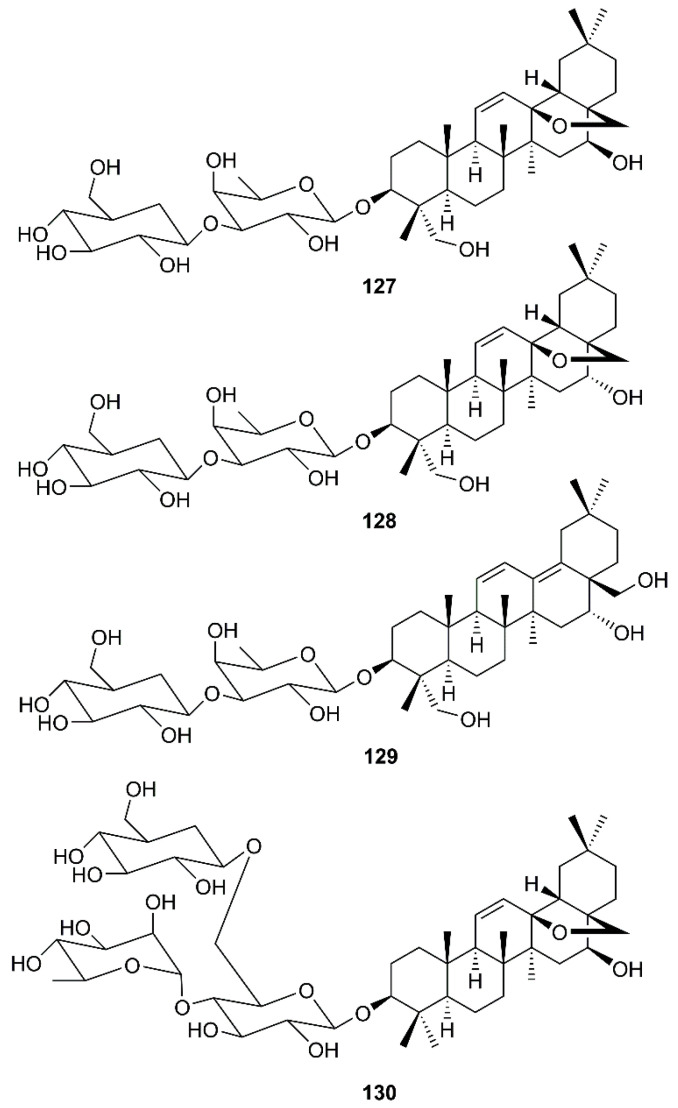
Structures of saikosaponins **127**–**130**.

**Figure 20 molecules-26-01754-f020:**
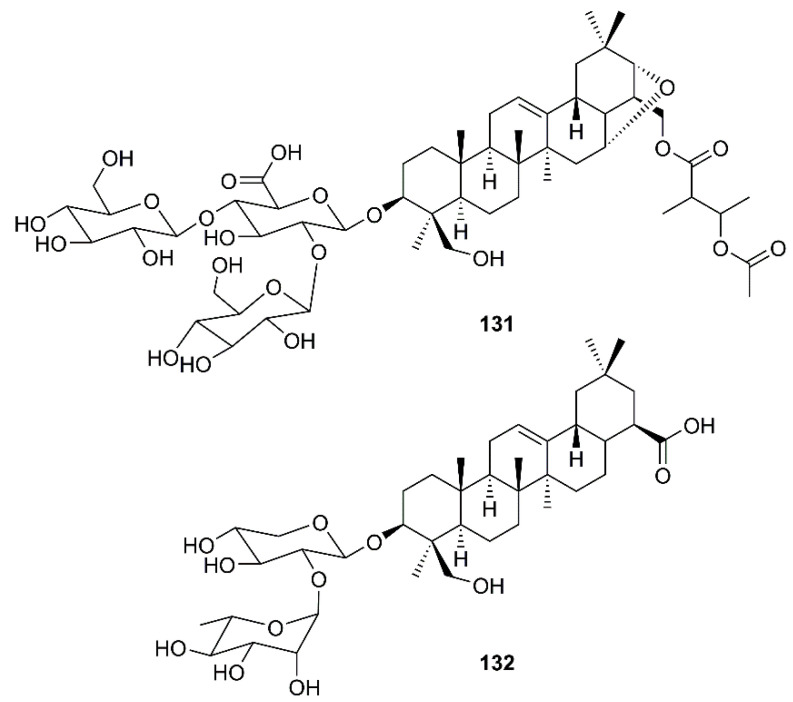
Structures of triterpene glycosides aescin (**131**) and α-hederin (**132**).

**Table 1 molecules-26-01754-t001:** Anti-Human Coronavirus (anti-HCoV) activities of some medicinal plants.

Plant Name	Family	Part Used	Ref.
*Toona sinensis* Reom.	Meliaceae	Leaves	[18]
*Cimicifuga racemosa* (L.) Nutt.	Ranunculaceae	Rhizome	[19]
*Melia azedarach* L.	Meliaceae	Barks	
*Coptis chinensis* Franch.	Ranunculaceae	Rhizomes	
*Phellodendron amurense* Rupr.	Rutaceae	Barks	
*Sophora subprostrata* Chun & T.Chen.	Fabaceae	Seeds	
*Paeonia suffruticosa* Andrews	Paeoniaceae	Whole plant	
*Houttuynia cordata* Thunb.	Saururaceae	Aerial parts	[20]
*Lycoris radiata* Herb.	Amaryllidaceae	Stem cortex	[23]
*Artemisia annua* L.	Asteraceae	Whole plant	
*Pyrrosia lingua* (Thunb.) Farw.	Ploypodiaceae	Leaves	
*Lindera aggregata* (Sims) Kostem	Lauraceae	Roots	
*Echinacea purpurea* (L.) Moench	Asteraceae	Aerial parts	[24]
*Anthemis hyalina* DC.	Asteraceae	Flowers	[26]
*Nigella sativa* L.	Ranunculaceae	Seeds	
*Citrus sinensis* L. Osbeck	Rutaceae	Peels	
*Astragalus mongholicus* Bunge	Fabaceae	Leaves	[29]
*Atractylodis macrocephalae* Koidz.	Asteraceae	Rhizome	
*Atractylodes lancea* (Thunb.) DC.	Asteraceae	Rhizome	
*Glycyrrhizae uralensis* Fisch. Ex DC.	Fabaceae	Roots	
*Saposhnikovia divaricata* (Turcz. Ex Ledeb.) Schischk.	Apiaceae	Flowers	
*Lonicerae japonica* Thunb.	Capripoliaceae	Fruits	
*Forsythia suspensa* (Thunb.) Vahl	Oleaceae	Leaves	
*Platycodon grandifiorus* (Jacq.) A.DC.	Campanulaceae	Roots	
*Agastache rugosa* (Fich. & C.A.Mey.) Kuntze	Lamiaceae	Aerial parts	
*Cyrtomium fortunei* J. Sm.	Dryopteridaceae	Leaves	
*Allium sativum* L. fam.	Alliaceae	Bulbs	[28]
*Camellia sinensis* (L.) Kuntze	Theaceae	Leaves	
*Zingiber officinalis*	Zingiberaceae	Roots	
*Hypericum perforatum* L.	Hypericaceae	Aerial parts	
*Scutellaria baicalensis* Georgi.	Lamiaceae	Aerial parts	[27]
*Bupleurum chinense* DC.	Apiaceae	Aerial parts	
*Gardenia jasminoides* J. Ellis.	Rubiaceae	Leaves	
*Stephania tetrandra* S. Moore	Menispermaceae	Leaves	[30]
*Rauwolfia* spp.	Apocynaceae	-	[31]
*Aesculus hippocastanum* L.	Sapindaceae	Seeds	
*Rheum officinale* Baill.	Polygonaceae	Roots	[32]
*Polygonum multiflorum* Thunb.	Polygonaceae	Roots	
*Scutellaria* spp.	Lamiaceae	Aerial parts	[33]
*Paulownia tomentosa* (Thunb.) Steud.	Paulowniaceae	Fruits	[34]
*Torreya nucifera* L. Siebold & Zucc.	Lamiaceae	Leaves	[35]
*Chamaecyparis obtusa* var. *formosana*	Cupressaceae	Heartwood	[36]
*Juniperus formosana* Hayata.	Cupressaceae	Heartwood	
*Cryptomeria japonica* (L.f.) D.Don	Cupressaceae	Heartwood	
*Rhus Chinensis* Mill.	Anacardiaceae	Fruits	[37]
*Laurus nobilis* L.	Lauraceae	Berry	[38]
*Thuja orientalis* L.	Cupressaceae	Fruits	
*Juniperus oxycedrus* ssp. *oxycedrus*	Cupressaceae	Berry	
*Euphorbia neriifolia* L.	Euphorbiaceae	Leaves	[39]
*Glycyrrhiza glabra* L.	Fabaceae	Roots	[40]
*Bupleurum* spp.	Apiaceae	Aerial parts	[41,42,43,44]
*Heteromorpha* spp.	Apiaceae	Aerial parts	
*Scrophularia scorodonia* L.	Scrophulariaceae	Aerial parts	

The same background color refers to the same reference(s).

**Table 2 molecules-26-01754-t002:** Anti-Human Coronavirus (Anti-HCoV) activities of natural compounds.

No.	Mode of Action	IC_50_ (μM)	CC_50_ (μM)	EC_50_ (μM)	SI	Concentration (μM)	Positive Control	Ref.
**Alkaloids**								
**7**	Inhibition of SARS-CoV (BJ-001 strain)	-	14,980.0 ± 912.0 ^a^	15.7 ± 1.2 ^a^	954	-	Interferon alpha, CC_50_ > 100,000 ± 710.1 μM, EC_50_ = 660.3 ± 119.1 μM, SI > 151	[23]
**8**	Inhibition of HCoV-OC43 infected MRC-5 human lung cells	0.33 ± 0.03	13.41 ± 0.36	-	40.19	2–20	-	[30]
**9**		1.01 ± 0.07	11.54 ± 0.46	-	11.46			
**10**		0.83 ± 0.07	11.26 ± 0.69	-	13.63			
**11**	Inhibition of SARS-CoV replication	-	25	3.4	7.3	-	-	[31]
**Anthraquinones**								
**18**	Blocks the interaction of SARS spike protein to ACE-2	200	-	-	-	0.1–400	Promazine	[32]
**Flavonoids and flavonoid glycosides**								
**19**	Inhibition of SARS-CoV 3CL Protease	8.3 (2.5 ± 0.8 μg/mL)	2718 (820 ± 15 μg/mL)	-	-	-	-	[47]
**22**	Inhibition of wild-type SARS-CoV infection	-	155	10.6 (9.2–12.2)	14.62	0.1–10,000	Glycyrrhizin (**111**), EC_50_ > 607.6 μM; Ribavirin, not effect	[44,50]
**23**	Inhibition of entry of HIV-luc/SARS pseudotypeed virus into Vero E6 cells	-	3320	83.4	39.81	0.1–10,000	-	[60]
**24**	Inhibition of SARS-CoV helicase, nsP13	2.71 ± 0.19	-	-	-	0.01–10		[51]
**25**		0.86 ± 0.48	-	-	-			
**31**	Inhibition of SARS-CoV 3CL Protease	33.17	-	-	-	2–320	-	[57]
**32**		27.45	-	-	-			
**33**		37.78	-	-	-			
**36**	Inhibition of SARS-CoV 3CL Protease	42.79 ± 4.97	-	-	-	9.4–80	-	[58]
**38**		24.14 ± 4.32	-	-	-			
**39**		31.62 ± 2.43	-	-	-			
**40**		48.85 ± 8.15	-	-	-			
**41**		61.46 ± 9.13	-	-	-			
**36**	Inhibition of SARS-CoV 3CL Protease Q189A	127.89 ± 10.06	-	-	-	16.5–200		
**45**	Inhibition of SARS-CoV papain-like protease	6.2 ± 0.04	-	-	-	0.1–100	-	[34]
**46**		6.1 ± 0.02	-	-	-			
**47**		11.6 ± 0.13	-	-	-			
**48**		12.5 ± 0.22	-	-	-			
**49**		5.0 ± 0.06	-	-	-			
**50**		9.5 ± 0.10	-	-	-			
**51**		9.2 ± 0.13	-	-	-			
**52**		13.2 ± 0.14	-	-	-			
**53**		12.7 ± 0.19	-	-	-			
**54**		14.4 ± 0.27	-	-	-			
**55**		10.4 ± 0.16	-	-	-			
**56**		13.9 ± 0.18	-	-	-			
**64**	Inhibition of SARS-CoV 3CL Protease	280.8 ± 21.4	-	-	-	1–1000	Luteolin (**22**), IC_50_ = 20.0 ± 2.2 μM; Quercetin (**23**), IC_50_ = 23.8 ± 1.9 μM	[35]
**65**		8.3 ± 1.2	-	-	-			
**66**		72.3 ± 4.5	-	-	-			
**67**		32.0 ± 1.7	-	-	-			
**68**		34.8 ± 0.2	-	-	-			
**Lignans and neolignans**								
**69**	Inhibition of Vero E6 cell proliferation and SARS-CoV replication	-	>750	>10	N.C. ^b^	0.01–10	Niclosamide, CC_50_ = 22.1 μM, EC_50_ < 0.1 μM, SI > 221; Valinomycin, CC_50_ = 67.5 μM, EC_50_ = 1.63 μM, SI = 41.4	[36]
**70**		-	>750	1.13	>667			
**72**		-	88.9	6.50	13.7			
**73**		-	68.3	3.80	18.0			
**69**	Inhibition of SARS-CoV 3CL Protease	>100	-	-	-	8–80	Niclosamide, IC_50_ = 40 μM	
**70**		25	-	-	-			
**Gallic acid derivatives**								
**74**	Inhibition of wild-type SARS-CoV infection	-	1.08	4.5 (1.96–5.8)	240	0.1–10,000	Glycyrrhizin (**111**), EC_50_ > 607.6 μM; Ribavirin, not effect	[60]
**75**	Inhibition of SARS-CoV 3CL Protease	3	-	-	-	4–20	*N*-Ethylmaleimide	[61]
**76**		43	-	-	-			
**77**		9.5	-	-	-			
**Sesquiterpenoids**								
**84**	Inhibition of Vero E6 cell proliferation and SARS-CoV replication	-	>750	>10	N.C. ^b^	0.01–10	Niclosamide, CC_50_ = 22.1 μM, EC_50_ < 0.1 μM, SI > 221; Valinomycin, CC_50_ = 67.5 μM, EC_50_ = 1.63 μM, SI = 41.4	[36]
**85**		-	76.8	4.44	17.3			
**Diterpenoids**								
**87**	Inhibition of Vero E6 cell proliferation and SARS-CoV replication	-	80.4	1.39	58.0	0.01–10	Niclosamide, CC_50_ = 22.1 μM, EC_50_ < 0.1 μM, SI > 221; Valinomycin, CC_50_ = 67.5 μM, EC_50_ = 1.63 μM, SI = 41.4	[36]
**88**		-	305.1	4.00	76.3			
**90**		-	78.5	>10	<7.9			
**91**		-	>750	1.47	>510			
**92**		-	127	1.15	111			
**93**		-	89.7	5.55	16.2			
**94**		-	303.3	1.57	193			
**95**		-	>750	4.71	>159			
**96**		-	674	7.5	89.8			
**87**	Inhibition of SARS-CoV 3CL Protease	49.6 ± 1.5	-	-	-	0.1–1000	Abietic acid (104), IC_50_ = 189.1 ± 15.5 μM	[35]
**97**		220.8 ± 10.4	-	-	-			
**98**		233.4 ± 22.2	-	-	-			
**99**		163.2 ± 13.8	-	-	-			
**100**		128.9 ± 25.2	-	-	-			
**101**		207.0 ± 14.3	-	-	-			
**102**		283.5 ± 18.4	-	-	-			
**103**		137.7 ± 12.5	-	-	-			
**Triterpenoids**								
**105**	Inhibition of Vero E6 cell proliferation and SARS-CoV replication	-	150	>10	<15	-	Niclosamide, CC_50_ = 22.1 μM, EC_50_ < 0.1 μM, SI > 221; Valinomycin, CC_50_ = 67.5 μM, EC_50_ = 1.63 μM, SI = 41.4	[36]
**106**		-	112	0.63	180			
**105**	Inhibition of SARS-CoV 3CL Protease	10	-	-	-	8–80	Niclosamide, IC_50_ = 40 μM	
**106**		>100	-	-	-			
**Saponins**								
**111**		-	>20,000 *	300 (51) *	>67	-	-	[40]
**112**	Inhibition of SARS-CoV replication	-	>3000	40 ± 13	>75	0.1–1000	Glycyrrhizin (**111**), CC_50_ > 24,000 μM, EC_50_ = 365. ± 12 μM, SI > 65	[63]
**113**		-	1462 ± 50	35 ± 7	41			
**114**		-	215 ± 18	139 ± 20	2			
**120**		-	44 ± 6	8 ± 2	6			
**121**		-	250 ± 19	50 ± 10	5			
**122**		-	15 ± 3	5 ± 3	3			
**123**		-	66 ± 8	16 ± 1	4			
**127**	Inhibition of HCoV-OC43 infected MRC-5 human lung cells	-	228.1 ± 3.8	8.6 ± 0.3	26.6	0.25–25	Actinomycin D, CC_50_ = 2.8 ± 0.3 μM, EC_50_ = 0.02 ± 0.0 μM, SI = 140	[64]
**128**		-	383.3 ± 0.2	1.7 ± 0.1 **	221.9			
**129**		-	121.5 ± 0.1	19.9 ± 0.1 *^†^	19.2			
**130**		-	176.2 ± 0.2	13.2 ± 0.3 *^†‡^	13.3			
**131**	Inhibition of SARS-CoV replication	-	15.0	6.0	2.5	-	-	[30]

^a^ nm; ^b^ not calculable; * mg/L; ** *p* < 0.05 compared with saikosaponin A; *^†^
*p* < 0.05 compared with saikosaponin B_2_; *^†‡^
*p* < 0.05 compared with saikosaponin C (student’s test).

## Data Availability

Data sharing is not applicable to this article.

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
