# Peer review of "Natural Products from Medicinal Plants with Anti-Human Coronavirus Activities"

_molecules, 2021, doi:10.3390/molecules26061754_

Round 1
Reviewer 1 Report
In the paper entitled “Natural Products from Medicinal Plants as Models for Anti-Coronavirus Drug Leads” the authors provide an overview of the plants and where available, the compounds isolated and tested from these plants for their antiviral activity against Corona viruses.
The paper is well-written, motivated and conceptualised. The paper could make valuable contribution to inform future research on treating Corona virus infection, but in its current form, it is not serving this purpose. There is very little interpretation and critical evaluation of the data presented. Furthermore, the methodology section is not included. What were the inclusion and exclusion criteria? How many papers were sourced? Where were these sources found etc.?
The paper is therefore slightly superficial for a scientific paper, as there is very little interpretation and conclusions drawn from the data. To summarise the information in literature is valuable, but a more in-depth analysis of the data is expected. In very few cases there is mention of a positive control, to compare the results obtained from the studies. How can the data be interpreted if it is not compared at least to a standard? In many sections there are mention of various assays and various viruses. How can these be compared? Any reflection on the type of assays and the results obtained? There is no mention of the ADME effects of compounds and the performance of compounds in an in-vivo situation. Results of many compounds are only based on molecular docking analysis, without further support for the results. Many of the compounds mentioned in the paper are well-known compounds and ADME effects/in-vivo performance of these compounds are available in literature. The presentation of the data can also be more effective. Most of the information in this manuscript would be more valuable in table format, as this will allow comparison of data. A column providing the positive control value and the comparison to the tested compounds would be more informative than a paragraph merely presenting which compounds have been tested before. More detailed comments are provided for each section.
Section 2 seems to be not comprehensive covering all plants. Why were these selected for this section? For section 2, a table should be included which comprehensively cover all plants/genus/families tested previously with anti-Corona virus activity. This should maybe include the distribution, the activity found, assay used, compounds identified etc.
Page 4 – Why the specific mention of these plants? There many plants used as remedies in traditional medicines. There is no reference to support the highlight of these plants?
“It is important to mention also that chamomile (Anthemis hyalina) flowers, black cumin (Nigella sativa) seeds, and Citrus sinensis (Rutaceae) peels, are commonly used as herbal remedies in traditional medicines in many cultures for treatment of a variety of human diseases”.
Section 3.1 – What is the commercially available sample? Why are there no mention of the results in comparison to a control? Are they better than the current treatments?
Section 3.2 – How does emodin compare to other compounds? Is it better or not? Toxicity? SI?
Section 3.3 – No controls? How effective are these compounds? For molecular docking, chloroquine is mentioned. Very little interpretation in this section. A number of compounds mentioned and the EC50 values are presented, but are these any better than what is available currently? There is no critical evaluation of the assays used. Most of these studies based on molecular docking, but there is no critical analysis on the use of molecular docking and the link to the in-vivo situation. No information on biotransformation or in-vivo effects considered. Are these worth mentioning if the compounds will be transformed in-vivo? Then these would not fit the molecular docking analysis?
Section 3.5 No interpretation and comparison to positive control.
Section 3.61 and 3.6.2 – Comparison to positive controls?
Section 6.3.4 – Control provided, but no concentration of compounds or the control to critically evaluate the potential of the compounds?
Section 3.6.5 – No controls. Which of these have potential. Solubility and in-vivo effects of the compounds? Different viruses and different assays. Can these results be compared?
Author Response
Reviewer #1’s comment: The paper is well-written, motivated and conceptualized. The paper could make valuable contribution to inform future research on treating Corona virus infection, but in its current form, it is not serving this purpose. There is very little interpretation and critical evaluation of the data presented. Furthermore, the methodology section is not included. What were the inclusion and exclusion criteria? How many papers were sourced? Where were these sources found etc.?
Reply: This manuscript is a review paper based on reports from the published papers of the medicinal plants extracts and their constituents that exhibited anti-human coronavirus activities and not a meta-analysis. Therefore reviewer#1’s suggestion to include “the methodology section and the inclusion and exclusion criteria” is not applied for this case. As reviewer#1 must have seen that all review papers (including some of the corresponding authors) published in Molecules have the same format and objectives.
Reviewer #1’s comment: The paper is therefore slightly superficial for a scientific paper, as there is very little interpretation and conclusions drawn from the data. To summarise the information in literature is valuable, but a more in-depth analysis of the data is expected. In very few cases there is mention of a positive control, to compare the results obtained from the studies. How can the data be interpreted if it is not compared at least to a standard? In many sections there are mention of various assays and various viruses. How can these be compared? Any reflection on the type of assays and the results obtained? There is no mention of the ADME effects of compounds and the performance of compounds in an in-vivo situation. Results of many compounds are only based on molecular docking analysis, without further support for the results. Many of the compounds mentioned in the paper are well-known compounds and ADME effects/in-vivo performance of these compounds are available in literature. The presentation of the data can also be more effective. Most of the information in this manuscript would be more valuable in table format, as this will allow comparison of data. A column providing the positive control value and the comparison to the tested compounds would be more informative than a paragraph merely presenting which compounds have been tested before. More detailed comments are provided for each section.
Reply: In our point of view and many years of doing research in the field of bioactive Natural Products and Medicinal Chemistry, we do agree that “ADME” is very important in research for drug development which involves pharmacology, pharmacokinetics and toxicology. However, this is not the objective of this manuscript. The purpose of this manuscript is to review the scientific-based evidences of some medicinal plants and their phytochemicals which are used in many countries for treatment of COVID-19 symptoms. As reviewer #1 can see from all the cited references that bioassays of the extracts and compounds have been carried out in vitro and neither of them in vivo since there is still no initiative to launch any drug development program from natural products. Anyhow, docking studies have been performed in many cases in an attempt to explain the demonstrated activity of some compounds present in the extracts. Therefore, in order to avoid misinterpretation of the manuscript title that might occur with someone who are not familiarized with this research field, we have modified the title of the manuscript to “Natural Products from Medicinal Plants with Anti-Human Coronavirus Activities”. We are convinced that this review will provide a vision to researchers, in particular Medicinal Chemists, to select particular scaffolds for modification to obtain compounds with improved activity and less toxicity as we have witnessed from several examples in this manuscript.
Reviewer #1’s comment: Section 2 seems to be not comprehensive covering all plants. Why were these selected for this section? For section 2, a table should be included which comprehensively cover all plants/genus/families tested previously with anti-Corona virus activity. This should maybe include the distribution, the activity found, assay used, compounds identified etc.
Reply: We wish to thank reviewer#1 for this valuable suggestion. Therefore, we have added Table 1 in Section 2 in the manuscript. Table 1 lists a number of medicinal plants, their families and part used as well as reference(s) for each plant. Moreover, we also added Table 2 which describes IC50, CC50, EC50, selective index (SI), concentration, positive control for the compounds mentioned in the text. We believe that these two tables will improve a readability of this manuscript.
Reviewer #1’s comment: Page 4 – Why the specific mention of these plants? There many plants used as remedies in traditional medicines. There is no reference to support the highlight of these plants?
“It is important to mention also that chamomile (Anthemis hyalina) flowers, black cumin (Nigella sativa) seeds, and Citrus sinensis (Rutaceae) peels, are commonly used as herbal remedies in traditional medicines in many cultures for treatment of a variety of human diseases”.
Reply: From a scientific point of view, there is no specific reason to mention chamomile (Anthemis hyalina) flowers, black cumin (Nigella sativa) seeds, and Citrus sinensis (Rutaceae) peels. However, it is understandable that as COVID-19 has emerged in all a sudden and spread so quickly, it is logical that many researchers have resorted common plant sources that have been used in many cultures for treatment of a variety of diseases to test for their capacity to protect against COVID-19. That is why chamomile flowers, black cumin seeds, and citrus peels have been widely investigated. Therefore, in our literature search, these plants have appeared as the most researched species.
Reviewer #1’s comment: Section 3.1 – What is the commercially available sample? Why are there no mention of the results in comparison to a control? Are they better than the current treatments?
Reply: The commercially available sample refers to a synthetic lycorine which is commercialized by Sigma-Aldrich or many firms in China such as NICBP in Beijing. Reviewer #1 can browse the link to see the details of the results from the reference Li et al. (https://doi.org/10.1016/j.antiviral.2005.02.007). We have also added the control (interferon alpha) in Table 2. For more details, please browse: https://doi.org/10.1016/s0140-6736(03)13973-6
Reviewer #1’s comment: Section 3.2 – How does emodin compare to other compounds? Is it better or not? Toxicity? SI?
Reply: From 312 controlled Chinese medicinal herbs, just three widely used Chinese medicinal plants of the family Polygonaceae contain emodin (18) that inhibited the interaction of the SARS-CoV S protein with ACE2. Compared to the positive control promazine, emodin (18) was also able to block the binding of the S protein of SARS-CoV to Vero E6 cells. The percent inhibition of promazine and emodin wwas 95.6 ± 7.7% and 94.12 ± 5.90% at concentration of 5 and 50 µM, respectively. These data clearly indicate that emodin is not as effective as promazine in inhibition of the S protein. However, it is not possible to compare the effectiveness of emodin (18) with other compounds as they were not assayed in the same conditions.
Reviewer #1’s comment: Section 3.3 – No controls? How effective are these compounds? For molecular docking, chloroquine is mentioned. Very little interpretation in this section. A number of compounds mentioned and the EC50 values are presented, but are these any better than what is available currently? There is no critical evaluation of the assays used. Most of these studies based on molecular docking, but there is no critical analysis on the use of molecular docking and the link to the in-vivo situation. No information on biotransformation or in-vivo effects considered. Are these worth mentioning if the compounds will be transformed in-vivo? Then these would not fit the molecular docking analysis?
Reply: Once again, reviewer#1 seems to require interpretation of the non-existing data. As stated above, there were no in vivo studies of extracts and compounds mentioned in this review since no references provided these data. This is understandable as most of the preparations used are based on extracts which were assayed for their anti-HCoV in vitro by several assay methods. These extracts were also analyzed, in most part, for their chemical constituents by HPLC/MS. I am not sure if reviewer#1 has any experience in working with isolation and structure elucidation of natural products. If so, reviewer#1 would understand that only a few mg amount of phytochemicals is normally isolated and purified from many grams of crude extracts. This quantity is only sufficient for structure elucidation and in vitro enzyme or cell-based assays. To be able to perform the in vivo study, it is necessary to obtain a good quantity, let’s say gram amount, of compounds. This quantity is not possible to obtain by isolation from natural sources but by synthesis. Therefore, a majority of bioassays of isolated pure phytochemicals are performed in vitro and not in vivo. That’s why the study of biotransformation of these naturally occurring phytochemicals is not a reality safe for commercially available or synthetic compounds.
As for chloroquine (and also ivermectin), it is more than proven that they have no activity. Concerning docking studies, there are a bloom of this method in an attempt to explain how these bioactive compounds work. Although we recognized its validity, what happens in living cells can be completely different from that coming from a computer simulation. Concerning this issue, and to guarantee a validity of the studies, we have replaced references regarding molecular docking in pre-prints with the published papers. (https://doi.org/10.3390/antiox9080742; https://doi.org/10.1016/j.phrs.2020.105255; https://doi.org/10.3390/molecules25173980).
Reviewer #1’s comment: Section 3.5. No interpretation and comparison to positive control.
Reply: By screening a natural product library consisting of 720 compounds for inhibitory activity against SARS-CoV 3CLPro, just two compounds in the library were found to possess inhibitory activity, i. e. tannic acid (75) and 3-isotheaflavin-3-gallate (TF2B; 76) with IC50 of 3 and 7 μM. Although, the reference from which we extracted these data mentioned the use of N-ethylmaleimide (NEM), a protease inhibitor, as positive control, its IC50 was not determined by HPLC assay. Therefore, we have added this statement in the manuscript.
Reviewer #1’s comment: Section 3.6.1 and 3.6.2 – Comparison to positive controls?
Reply: For Section 3.6.1, the positive control was glycyrrhizin (compound 111), which is mentioned in the text in the manuscript. For section 3.6.2, the positive controls were niclosamide and valinomycin, which are mentioned in the Table 2.
Reviewer #1’s comment: Section 6.3.4 – Control provided, but no concentration of compounds or the control to critically evaluate the potential of the compounds?
Reply: The positive controls were niclosamide and valinomycin which are also mentioned in the Table 2.
The positive control for fridelane triterpenoids was actinomycin D which is also mentioned in Table 2.
Reviewer #1’s comment: Section 3.6.5 – No controls. Which of these have potential. Solubility and in-vivo effects of the compounds? Different viruses and different assays. Can these results be compared?
Reply: The positive control was glycyrrhizin (compound 111). In relation to solubility and in-vivo effects of the compounds, it is well known that a majority of small molecules are not water soluble. However, as the quantity of the compounds present in the extract is very small, its solubility changed when they are present as a mixture or in a preparation. As reviewer#1 must be aware that normally different assays are used to determine biological activity of compounds, and new assays with advanced techniques and more accuracy are emerging every year. However, this cannot invalidate the results obtained from different assays. Moreover, it is not our intention to compare the assay methods used in determination of the activities of the compounds.
Reviewer 2 Report
It is an excellent manuscript that of Dr. Salar Hafez Ghoran et al., Carried out a good bibliographic review
I only suggest that you be more specific about the possible mechanisms of action of these plants and their metabolites.
It is known that Covid causes inflammation, oxidative stress and a cascade of cytokines, can they make a better explanation about how plants interact?
Can you comment on what is known about the regulation of the transcriptional factor NRF2?
Can you make figures where you can see how they protect these plants from covid?
Author Response
Reviewer #2:
It is an excellent manuscript that of Dr. Salar Hafez Ghoran et al., Carried out a good bibliographic review
Reply: We sincerely appreciate reviewer#2’s opinion about the manuscript.
Reviewer #2: I only suggest that you be more specific about the possible mechanisms of action of these plants and their metabolites. It is known that Covid causes inflammation, oxidative stress and a cascade of cytokines, can they make a better explanation about how plants interact?
Reply: We deeply thank reviewer#2’s valuable suggestion. We have followed reviewer#2’s suggestion by adding a brief mention of the inflammation and cytokine storm in the Introduction.
Reviewer #2: Can you comment on what is known about the regulation of the transcriptional factor NRF2?
Reply: As our objective is to focus on medicinal plants and their phytochemicals possessing activities against SARS-CoV, we haven’t gone further afield into the domain of immunology. However, it is undeniable that this is another interesting subject to be explored concerning COVID-19 issue.
Reviewer #2: Can you make figures where you can see how they protect these plants from covid?
Reply: If we understand correctly, reviewer#2 asked if we can draw figures to show how these plants protect us from COVID-19. If so, we would say that it is very difficult to draw figures for this purpose as different mechanisms are operating in the process of SARS-CoV-2 infection and much of the details is still unknown. This is a very complicated process and thus it is not easy to show it by figures.
Reviewer 3 Report
The manuscript molecules-1139825 presents a wide review about plants and their metabolites that already have some suggestion or indication of anti-viral activity so that some molecule can be suggested for further tests using SARS-CoV-2.
The text is dense, has good writing (although it needs a revision in some terms in English), but in many moments the text becomes tiresome and makes the reader lose the focus of the text easily.
In addition, some referenced papers have been in preprint form for over 1 year without ever having been properly peer-reviewed. I pointed out in the text one of these that brings doubtful information and that performed an in silico test in the wrong way.
Thus, I suggest that the authors use in the review only papers published in quality journals and that have been reviewed by peers.
After these changes and adjustments and after a new review, the manuscript can be accepted for publication.

Author Response
Reviewer #3:
The manuscript molecules-1139825 presents a wide review about plants and their metabolites that already have some suggestion or indication of anti-viral activity so that some molecule can be suggested for further tests using SARS-CoV-2.
The text is dense, has good writing (although it needs a revision in some terms in English), but in many moments the text becomes tiresome and makes the reader lose the focus of the text easily.
In addition, some referenced papers have been in preprint form for over 1 year without ever having been properly peer-reviewed. I pointed out in the text one of these that brings doubtful information and that performed an in silico test in the wrong way. Thus, I suggest that the authors use in the review only papers published in quality journals and that have been reviewed by peers.
After these changes and adjustments and after a new review, the manuscript can be accepted for publication.
Reply: We would like to thank reviewer#3 for constructive comments and suggestions. Therefore we have done the following:
- The word “avenues” was replaced by “approaches”.
- For “anti-adhesive”, another definition has been used.
- All plant names were carefully rechecked and the names of identifiers were added. In addition, we have added Table 1 which provided plant name, family, and the part used for all the plants mentioned in the text.
- The PCA analysis suggestion is really importance, however, it is beyond our expertise and the scope of this review.
- The missed “μ” was added for “1 to 10 μg/mL”.
- The preprints references were removed and the published and credible references were used where needed.
Reviewer 4 Report
The review entitled “Natural Products from Medicinal Plants as Models for Anti-Coronavirus Drug Leads” by Ghoran et al., is well written and articulated.
This is an intensive chemical biology paper on anti-coronavirus drugs. Although a lot more opinions/reviews have been published, still there is a space for papers that discuss strategies against coronavirus. Besides, his detailed explanation of SAR based structure designs would be useful for future research.
Here are some minor comments
The manuscript deserves extensive English corrections.
Introduction
Expand the SARS at first instance.
The term “COVID” should be uniform throughout the manuscript.
A brief introduction about the various strategies that target SARS-CoV-2 and COVID-19 should be presented. I hope these references would be useful… https://doi.org/10.1016/j.nantod.2020.101051; https://doi.org/10.1016/j.xcrm.2020.100016.
Page 4: Nucleocapsid (N) and Spike (S) proteins
As the text is very comprehensive, I would recommend authors to prepare a table, which segregated into the class of compounds, type of virus, IC50 values, mechanism of action, etc.
Authors could be benefited from the following reference https://doi.org/10.3389/fimmu.2020.586572.
Author Response
Reviewer #4:
The review entitled “Natural Products from Medicinal Plants as Models for Anti-Coronavirus Drug Leads” by Ghoran et al., is well written and articulated.
This is an intensive chemical biology paper on anti-coronavirus drugs. Although a lot more opinions/reviews have been published, still there is a space for papers that discuss strategies against coronavirus. Besides, his detailed explanation of SAR based structure designs would be useful for future research.
Here are some minor comments:
Reviewer #4: The manuscript deserves extensive English corrections.
Reply: The manuscript has been carefully rechecked and the English language is improved.
Introduction
Reviewer #4: Expand the SARS at first instance.
Reply: The SARS expansion has been already mentioned in the line 5.
Reviewer #4: The term “COVID” should be uniform throughout the manuscript.
Reply: The term “COVID” was put in uniform throughout the manuscript.
Reviewer #4: A brief introduction about the various strategies that target SARS-CoV-2 and COVID-19 should be presented. I hope these references would be useful… https://doi.org/10.1016/j.nantod.2020.101051; https://doi.org/10.1016/j.xcrm.2020.100016.
Reply: We wish to thank reviewer #4’s valuable suggestions. We have added a paragraph
explaining various strategies that target SARS-CoV-2 and COVID-19 in the Introduction.
Reviewer #4: Page 4: Nucleocapsid (N) and Spike (S) proteins.
Reply: The expand terms of N and S were added.
Reviewer #4: As the text is very comprehensive, I would recommend authors to prepare a table, which segregated into the class of compounds, type of virus, IC50 values, mechanism of action, etc.
Authors could be benefited from the following reference https://doi.org/10.3389/fimmu.2020.586572
Reply: We thank reviewer #4 again for his/her suggestion. We have added Tables 1 and 2 in the text. We are certain that these Tables will improve a readability of the manuscript.
Round 2
Reviewer 3 Report
The manuscript has been substantially improved both on a technical and scientific level.
I have no further thoughts on the present study.
My judgment is favorable to publication